EMBO
Molecular Medicine

# Saccharin disrupts bacterial cell envelope stability and interferes with DNA replication dynamics

Rubén de Dios [1], Kavita Gadar [1], Chris R Proctor[1], Evgenia Maslova[1], Jie Han [1],
Mohamed A N Soliman [1], Dominika Krawiel[1], Emma L Dunbar[2], Bhupender Singh[3], Stelinda Peros[4],
Tom Killelea[5], Anna-Luisa Warnke [6], Marius M Haugland[6], Edward L Bolt[5], Christian S Lentz[3],
Christian J Rudolph [4] & Ronan R McCarthy [1]✉

## Abstract

**Saccharin has been part of the human diet for over 100 years, and there is a comprehensive body of evidence demonstrating that it can influence the gut microbiome, ultimately impacting human health. However, the precise mechanisms through which saccharin can impact bacteria have remained elusive. In this work, we demonstrate that saccharin inhibits cell division, leading to cell filamentation with altered DNA synthesis dynamics. We show that these effects on the cell are superseded by the formation of bulges emerging from the cell envelope, which ultimately trigger cell lysis. We demonstrate that saccharin can inhibit the growth of both Gram-negative and Gram-positive bacteria as well as disrupt key phenotypes linked to host colonisation, such as motility and biofilm formation. In addition, we test its potential to disrupt established biofilms (single-species as well as polymicrobial) and its capacity to re-sensitise multidrug-resistant pathogens to last-resort antibiotics. Finally, we present in vitro and ex vivo evidence of the versatility of saccharin as a potential antimicrobial by integrating it into an effective hydrogel wound dressing.**

**Keywords** Artificial Sweetener; *Acinetobacter baumannii*; Antimicrobial; Biofilm; DNA Replication
**Subject Category** Microbiology, Virology & Host Pathogen Interaction

## Introduction

The critical rise in global obesity has led to a progressive increase in the consumption of non-caloric artificial sweeteners worldwide. Currently, it is estimated that 25% of children and 41% of adults in the United States are daily consumers of artificial sweeteners (Sylvetsky et al, 2017). Among the most popular artificial sweeteners, saccharin has become the market leader in relation to its sweetening power (Sylvetsky and Rother, 2016). Saccharin ($C_7H_5NO_3S$) is a heat-stable artificial sweetener that is ~300–700 times sweeter than sucrose. In contrast, it has a zero net caloric contribution to the diet. It was first discovered by accident in the late 1870s by a chemist, Constantin Fahlberg, who noted an intense sweet taste on his hand while working on the development of benzoic sulfimide coal tar derivatives (Sylvetsky and Rother, 2016). The sweetener was quickly commercialised and grew in popularity, particularly during World War I, when traditional sugar was in short supply. Its proliferation into the human diet was further accelerated in the 1960s, as it was marketed as a product to support weight loss.

The impact of artificial sweeteners on the host microbiome has become an emergent area of focus. The effect of saccharin on the gut microbiome was first reported in 1980, when male rats were fed 7.5% saccharin for 10 days. Although it showed no impact on the total anaerobe numbers in the caecum, specific sub-populations of anaerobes were depleted (Anderson and Kirkland, 1980). More recently, Suez et al (2014) established a link between saccharin consumption and glucose intolerance via alterations in the gut microbiota (Suez et al, 2014). However, a later study exploring the effects of saccharin on the gut microbiota and glucose tolerance in healthy mice and humans demonstrated no impact on the glucose or hormonal responses. No major impacts on the microbiome composition of either mice or humans were reported, although some minor shifts at the genus level were noted in mice (Serrano et al, 2021). Interestingly, while several additional studies have pointed towards an impact of sweeteners, including saccharin, on the microbiome, there appears to be a lack of a consistent signature across these studies that could be used as a marker of artificial sweetener consumption. Nevertheless, it has been reported that artificial sweeteners, including saccharin, can trigger inflammatory responses by different means. Bian et al (2017) gave saccharin to mice via drinking water (0.3 mg/ml) and found that this induced

[1]Antimicrobial Innovations Centre, Division of Biosciences, Department of Life Sciences, College of Health, Medicine and Life Sciences, Brunel University London, Uxbridge UB8 3PH, UK. [2]Department of Biochemistry, University of Wisconsin-Madison, Madison, WI 53706-1544, USA. [3]Research Group for Host-Microbe Interactions, Department of Medical Biology and Centre for New Antibacterial Strategies (CANS), UiT—The Arctic University of Norway, 9019 Tromsø, Norway. [4]Division of Biosciences, Department of Life Sciences, Centre for Genome Engineering and Maintenance, College of Health, Medicine and Life Sciences, Brunel University London, Uxbridge UB8 3PH, UK. [5]School of Life Sciences, Faculty of Medicine & Health Sciences, Queens Medical Centre, University of Nottingham, Nottingham NG7 2UH, UK. [6]Department of Chemistry, UiT—The Arctic University of Norway, 9037 Tromsø, Norway. ✉E-mail: ronan.mccarthy@brunel.ac.uk

gut microbiome alterations that coincided with an increase in pro-inflammatory mediators. Moreover, Skurk et al (2023) demonstrated that saccharin, at similar concentrations to those expected in plasma after dietary intake, produced alterations in the transcriptional signature of neutrophils. This led to a shift in their state from "homoeostasis" to "priming", thus promoting inflammation. In contrast to this, other reports have shown that saccharin derivatives can inhibit other immune pathways leading to inflammation and have even been proposed as antagonists to modulate interferon-mediated inflammation (Csakai et al, 2014). This points towards a highly specific effect of saccharin on the microbiome, but a multifactorial effect on the host, specifically on the immune system, which would altogether be heavily influenced by the host diet and lifestyle.

The effect at the microbiome level has received much attention, but the impact of saccharin on specific bacterial species at the cellular level is less clear. However, there is a mounting body of evidence that saccharin can influence bacterial growth, including an inhibition of members of the oral microbiome, such as *Porphyromonas gingivalis* and *Aggregatibacter actinomycetemcomitans* (Prashant et al, 2012). Similarly, it has been shown to limit the growth of a range of lab-model bacteria including *Staphylococcus aureus*, *Bacillus cereus*, *Klebsiella pneumoniae* and *Pseudomonas aeruginosa* (Sünderhauf et al, 2020; Wang et al, 2018). Furthermore, saccharin has been shown to influence natural transformation in a range of environmental bacterial species (Yu et al, 2021; Yu et al, 2022). Recent work has also demonstrated that saccharin, along with other artificial sweeteners such as acesulfame-K (ace-K), can inhibit the growth of multidrug-resistant (MDR) *P. aeruginosa* and *Acinetobacter baumannii* (de Dios et al, 2023), the later occupying the first position on the World Health Organisation (WHO) priority pathogen list (Tacconelli et al, 2018).

Despite this mounting body of evidence, the mechanisms underpinning saccharin-mediated growth inhibition in bacteria remain unexplored. In this work, we provide direct evidence for the first time that saccharin can produce bulge-mediated cell lysis leading to cell death and alter DNA synthesis dynamics within the cell. We provide proof-of-concept results for the use of saccharin as an antimicrobial and anti-virulence therapeutic in a range of MDR bacteria. Apart from the direct antimicrobial effect of this artificial sweetener, we demonstrate that saccharin disrupts the cell envelope as a barrier. This facilitates a greater antibiotic penetration and overwhelms native resistance mechanisms in MDR *A. baumannii*, thus re-sensitising it to frontline antibiotics. As a final step, we demonstrate the therapeutic potential of saccharin as an antimicrobial in in vitro and ex vivo models.

# Results

## Saccharin induces filamentation and bulge-mediated cell lysis

Saccharin has previously been reported to have antimicrobial activity (Sünderhauf et al, 2020; Yu et al, 2022; Yu and Guo, 2022; de Dios et al, 2023), although the mechanisms underpinning this activity have remained elusive. To understand how the cell responds to saccharin exposure, we performed time-lapse microscopy using an *E. coli* model and treating with 1.4% saccharin, an effective concentration tested empirically in this setting and close to the theoretical half-maximal inhibitory concentration (IC50) (Fig. 1A; Appendix Fig. S1). In this setup, we could see that cells acquired an aberrant morphology, filamenting and swelling in the central section. As the treatment progresses, membrane bulges also appear and continue to grow. Eventually, this led to cell lysis, with the concomitant emergence of a "ghost" cell (Movies EV1 and EV2). This bulge-mediated cell lysis is remarkably similar to the morphological response we previously reported for cells when exposed to the sweetener ace-K (de Dios et al, 2023), suggesting a similar mechanism of action. Using the cardiolipin (CL)-specific fluorescent dye 10-N-nonyl-acridine orange (NAO) to visualise CL distribution, we could see clear structural rearrangements in the cell membrane, which align with previous reports of saccharin altering the membrane integrity and permeability (Yu et al, 2021, 2022; Yu and Guo, 2022), and confirm that the bulges were emerging from cells prior to lysis (Mileykovskaya and Dowhan, 2000).

To gain more insights on how saccharin may interfere with cell division, we performed differential staining at multiple timepoints after saccharin treatment with an *E. coli* strain bearing an eCFP-labelled version of *ftsZ*. This allowed us to track the progression of the membrane status over time, as well as septation (Fig. 1B,C). The membrane staining re-confirmed the loss of morphological integrity and the formation of bulges after saccharin treatment. FtsZ rings could clearly be seen forming in the cells indicating that the impact of saccharin on cell division was not due to the inhibition of septum formation but could be due to these septa being prevented or blocked from completing fission.

## Saccharin targets multiple essential cell processes, including DNA replication and repair

To further elucidate the effect of saccharin on *E. coli* and gain a greater insight into the cause of the aberrant morphologies and cell division phenotypes, we performed a differential RNA sequencing (dRNA-seq) experiment, comparing cultures exposed to 1.4% saccharin for 1 h to a mock treatment control. This analysis identified a total of 724 genes that were differentially regulated (Dataset EV1), with 419 significantly downregulated greater than |LogFC|>1 and 305 genes significantly upregulated greater than |LogFC|>1. Within the dRNA-seq dataset, we identified several membrane-associated genes being differentially regulated after saccharin exposure, which is concomitant with our observed effect on cell morphology. Specifically, the most downregulated gene was the outer membrane porin OmpF, while pathways associated with O-antigen LPS biosynthesis and peptidoglycan biosynthesis were significantly upregulated. This suggests a cellular response to cell envelope damage triggered by saccharin. Intriguingly, a KEGG pathway analysis showed that β-lactam resistance mechanisms were significantly upregulated (Dataset EV2). Indeed, the impact on the cellular morphology of saccharin is remarkably similar to that of peptidoglycan-targeting β-lactam antibiotics, with cellular filamentation and bulges forming on the cell membrane followed by cell lysis (Yao et al, 2012). This suggests the cell may be responding to saccharin in a similar manner as to how it responds to antibiotic exposure, in particular to the β-lactam class.

A striking observation revealed by the KEGG pathway analysis was that amongst the most significantly upregulated pathways were the DNA replication and mismatch repair systems. This suggests a

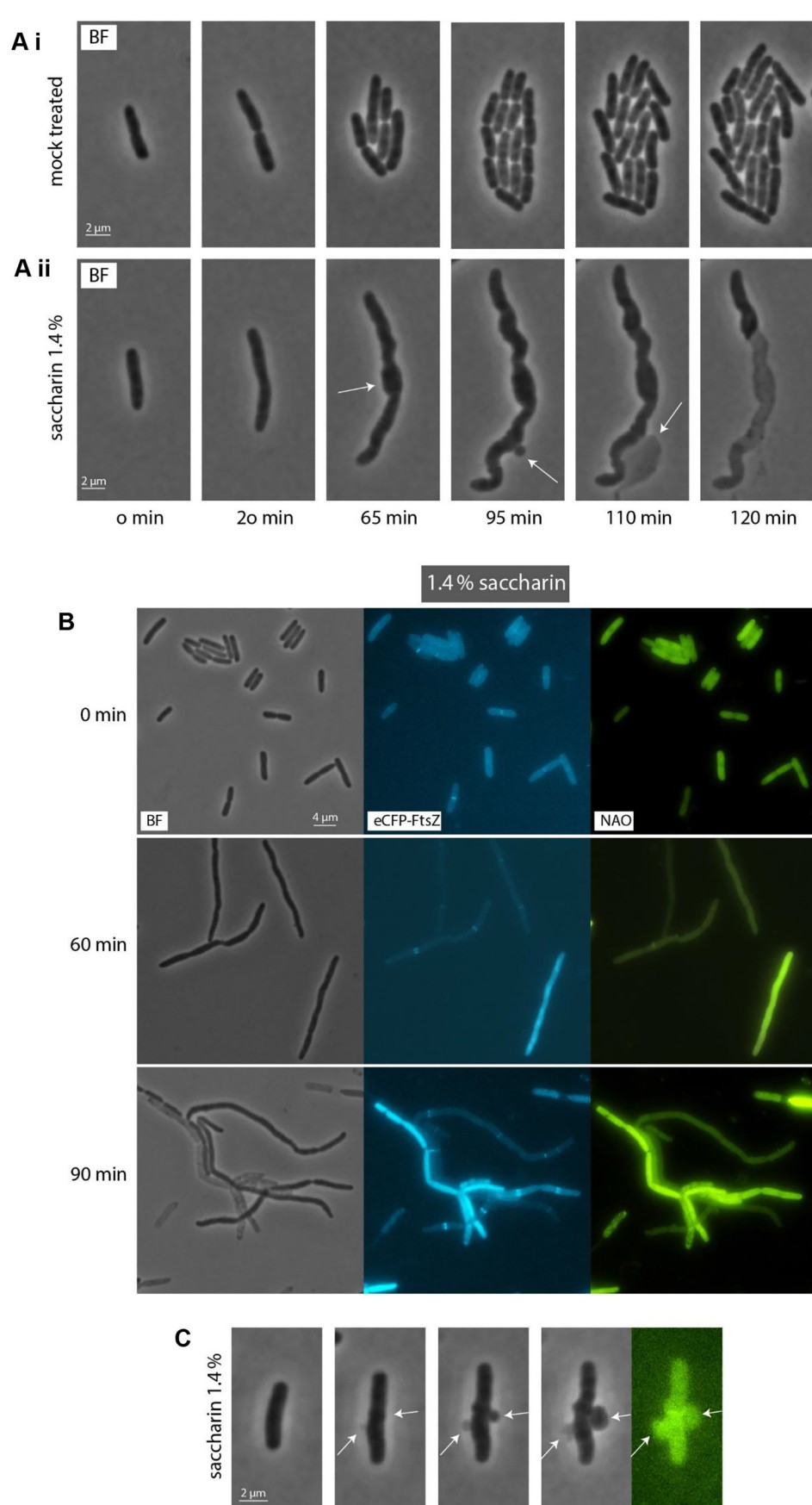

**Figure 1.  Impact of saccharin on cellular growth and morphology.**

(Ai–ii) Phase contrast images of a time lapse of *E. coli* MG1655 cells mock-treated (i) and treated with 1.4% saccharin (ii). An environmental chamber was used to maintain a constant temperature of 37 °C, and cells were imaged for 120 min. The images shown highlight the formation of a membrane bulge, which quickly becomes so extensive that it ruptures, leaving a "ghost" cell behind. (B) Phase contrast and fluorescent images of *E. coli* cells following treatment with saccharin. Cells were grown in LB broth (Miller) to the early exponential growth phase. Aliquots were transferred to pre-warmed culture tubes, and saccharin was added at the concentration indicated. Samples were taken at the time shown. The membrane was visualised by staining with 10-nonyl acridine orange (NAO) for 5 min before visualisation (see "Methods" for details). Z-ring formation was visualised by expressing a FtsZ allele fused to CFP (see "Methods" for details). The strain used was JD1709. (C) Time lapse of 1.4% saccharin-induced cell envelope bulging, with NAO staining the bulge. The images shown are representative examples from a minimum of three biological replicates. White arrows indicate cell regions experiencing membrane bulging. Source data are available online for this figure.

previously unreported effect of saccharin on bacterial genome dynamics. For this reason, we sought to explore the effect of saccharin on *E. coli* by monitoring the DNA replication dynamics in the cell during exposure to this artificial sweetener.

## Saccharin interferes with DNA replication dynamics

The primary source of DNA replication under physiological conditions is chromosomal replication from the origin of replication region (*ori*), forming two replisomes that move through the chromosome in opposite directions until fusing within the termination region (*ter*). To explore a potential impact on *ori*-initiated DNA synthesis following 1.4% saccharin exposure, we treated an *E. coli* strain in which the *ori* and *ter* regions of the chromosome can be tracked by fluorescent repressor-operator systems (FROS; Fig. 2A). Figure 2B shows the distribution of *ori* and *ter* foci corresponding to each documented cell after 60 min treatment. In the mock-treated control, we can observe the expected 2–4 *ori* foci, with the *ter* foci lagging one step behind in the duplication progression (1–2 foci). However, after 1.4% saccharin treatment, there is an amplification of both *ori* and *ter* foci, with the distribution of *ori* foci shifting towards 8–16 counts and beyond. The *ter* foci follow a similar trend but lagging behind in the duplication progression, as would be expected.

One possible explanation for this observed effect would be that the lack of cell division caused by saccharin may lead to an accumulation of replicating chromosomes in the filamented cells, in the same replicative state as though they had successfully accomplished division. Mock-treated cells which are able to grow normally show 2–4 origin foci and 1–2 terminus foci. Cell division in *E. coli* occurs every 20 min. Thus, within our experimental window of 60 min, cells can undergo three division events as an absolute maximum. However, as cultures are asynchronous, we would expect the majority of cells to have undergone 2 division events. If cell division is inhibited, we would therefore expect 8–16 *ori* foci and 4–8 *ter* foci, which aligns very well with the observed distributions observed after 60 min of saccharin treatment (Fig. 2B). This suggests that *ori*-initiated chromosomal replication is not significantly impacted by saccharin, but that multiple chromosomes undergoing replication accumulate in a saccharin-induced filamented cell.

However, *ori*-initiated replication is not the only possible source of DNA synthesis in the cell that could be potentially impacted by saccharin. We therefore wanted to test if saccharin may affect DNA synthesis initiated at sites other than the *ori*, in line with the results of our dRNA-seq experiment. We first tested the effect of saccharin on an *E. coli* strain carrying a DnaN fluorescent reporter (YPet-

DnaN, Fig. 2C,D). DnaN is the sliding clamp which allows direct visualisation of locations of active DNA replication. These results showed a significant increase in DnaN foci in cells treated with 1.4% saccharin compared to a mock control. To further distinguish if the DNA replication triggered by saccharin was started at the *ori* or away from these regions, we tested our YPet-DnaN reporter in an *E. coli* strain encoding a thermosensitive version of the chromosomal *ori*-dependent replication initiation protein DnaA (*dnaA*(ts)) and carrying a fluorescent DnaN reporter. In this strain, origin firing ceases upon a shift to 42 °C, while all ongoing rounds of synthesis are able to continue their journey until they terminate (Rudolph et al, 2009). In mock-treated cells, we could observe a shift of DnaN foci to either lower numbers or no foci at all at this restrictive temperature compared to 30 °C, as expected. However, cells treated with 1.4% saccharin showed an increase in DnaN foci numbers, both at 30 °C and 42 °C, compared to the mock-treated cells, which is in line with the idea that saccharin may trigger DNA synthesis away from the *ori*, either directly or indirectly (Appendix Fig. S2).

To enable the quick screening of a variety of existing mutant strains, we switched to an *E. coli* strain over-expressing a functional Cas1-Cas2 complex where Cas1 contains a linker, followed by an enhanced yellow fluorescent protein (eYFP). We will call this Cas1-linker-eYFP-Cas2 complex Cas1-Cas2 in the rest of the text, for simplicity. It was recently shown that Cas1-Cas2 localises into distinct foci in cells where DNA is being actively synthesised, allowing its use as a biomarker of active DNA replication (Killelea et al, 2023). This system is added simply via an expression plasmid, allowing the rapid generation of various strains. Using this reporter system, we could see that a mock-treated population would be mostly represented by cells with 1–2 Cas1-Cas2 foci, with part of the population shifted towards 4 foci. However, the distribution of cells dramatically shifted towards a population with increased Cas1-Cas2 foci numbers with saccharin treatment (Fig. 2E,F). As a control, we used a Cas1[R84G] derivative, which is unable to bind DNA. We showed before that Cas1[R84G]-Cas2 complexes showed foci only in small number of cells, and mostly in an aberrant location (Killelea et al, 2023). We did not observe any increase in the number of foci following saccharin treatment, highlighting the specificity of this effect (Appendix Fig. S3).

For both Cas1-Cas2 and fluorescently labelled DnaN, the majority of the cells contained 2–4 foci. As described above, most cells will undergo 2 division events within the 60 min treatment window. We would therefore expect most cells to show 8–16 foci, which is exactly what we observed, highlighting that a significant part of the signal accumulation is simply normal DNA synthesis in cells that fail to divide. However, in both experiments we did

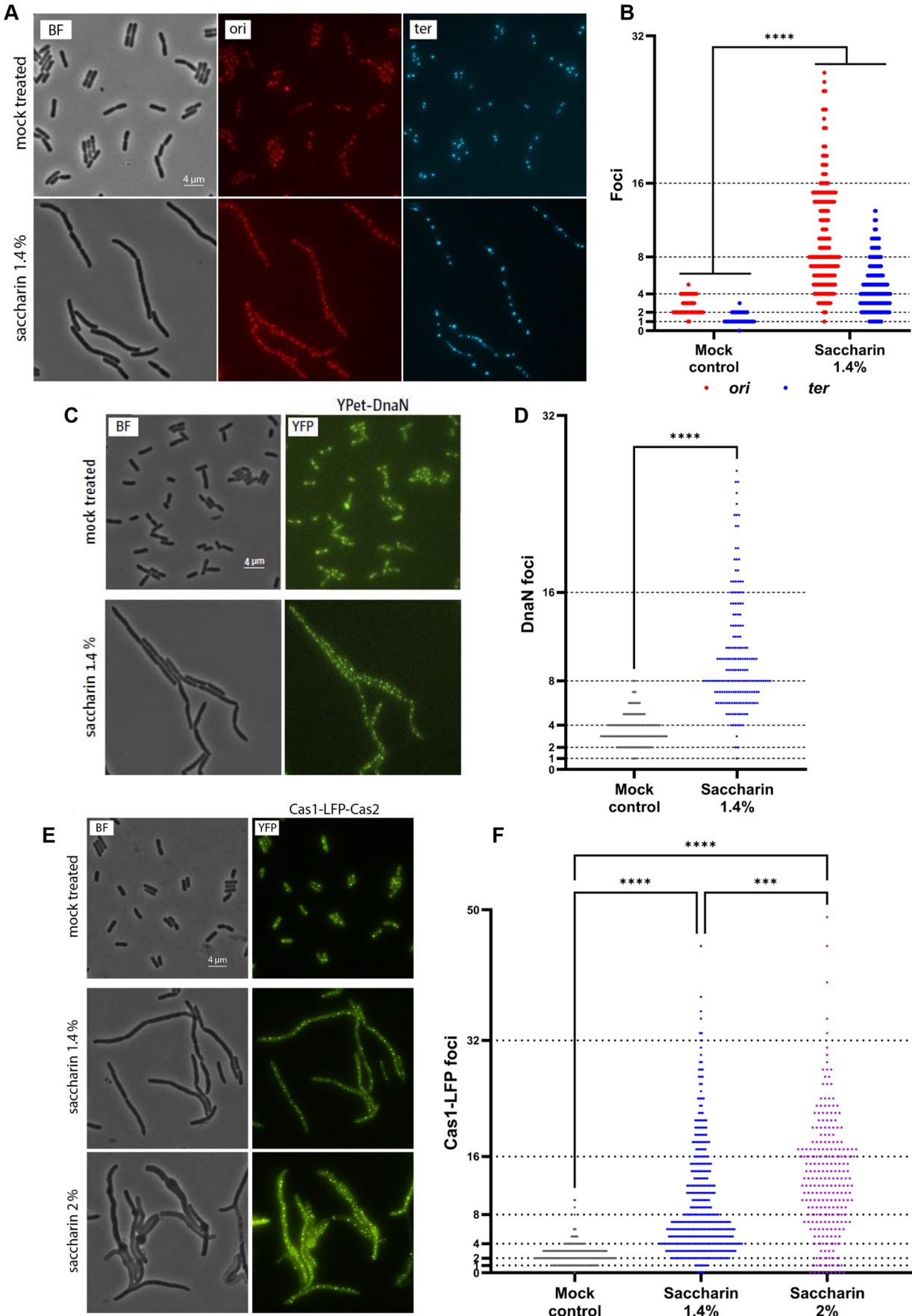

**Figure 2.  Replication dynamics in living cells later dramatically following treatment with saccharin.**

(A) The origin and terminus area of the chromosome (*ori* and *ter*, respectively) can be visualised via fluorescent repressor-operator arrays (FROS). The origin region is shown in red, whereas the terminus region is shown in cyan. Cells were grown to early exponential growth phase in LB broth (Miller). Untreated cells show foci numbers observed with growth conditions that allow overlapping rounds of DNA synthesis, resulting typically in 2–4 *ori* foci and 1–2 *ter* foci per cell. Treatment with saccharin results in a significant increase in the number of both origin and terminus foci, indicating amplification of both the origin and terminus areas in treated cells. The images shown are representative examples of three biological replicates. The strain used was RRL189. (B) Quantification of foci numbers from the experiments shown in (A). Dashed lines indicate the theoretically possible foci numbers expected from a normal duplication progression. Number of cells analysed, pooled from three biological replicates, is 470 for the mock-treated samples and 286 for the saccharin-treated samples. The results shown were analysed by the Mann–Whitney test between treated and control samples for *ori* and *ter* foci. (C) To visualise active DNA synthesis following saccharin treatment, we used a strain in which the β sliding clamp, DnaN, is fused to YPet, a brighter YFP derivative. *E. coli* MG1655 cells were grown to an early exponential growth phase in LB broth (Miller). Following treatment with saccharin, the formed filamentous cells show a significant increase in the number of YPet-DnaN foci. The strain used was AS1062. (D) Quantification of DnaN foci shown in (C). Shown are foci distributions highlighting the concentration-dependent increase of foci following treatment. The number of cells analysed in (C), pooled from three biological replicates, is 311 for the mock-treated samples and 176 for the samples treated with 1.4% saccharin. Data in (D) were analysed by Mann–Whitney test comparing treated versus mock-treated groups. (E) *E. coli* MG1655 cells were grown to early exponential growth phase in LB broth (Miller), and expression of Cas1-linker-eYFP and Cas2 (called Cas1-Cas2 for simplicity) from plasmid pTK135 induced by the addition of 0.1% arabinose for 60 min before visualisation. As shown recently (Killelea et al, 2023), active DNA replication results in robust focus formation of Cas1-Cas2, resulting in the majority of cells showing between 1 and 4 foci per cell. Following treatment with saccharin, this pattern is dramatically altered, with cells showing significantly higher foci numbers. The images shown are representative examples from a minimum of three biological replicates. The strain used was JD1708. (F) Quantification of Cas1-Cas2 foci shown in (E). Shown are foci distributions highlighting the concentration-dependent increase of foci following treatment. Dashed lines indicate the theoretically possible foci numbers expected from a normal duplication progression. The number of cells analysed in (F), pooled from three biological replicates, is 603 for the mock-treated samples, 414 for the samples treated with 1.4% saccharin and 241 for the samples treated with 2% saccharin. Results shown in (F) were analysed by the Kruskal–Wallis test with Dunn's correction. Significance is indicated as ***$P \le 0.001$, ****$P \le 0.0001$ (exact $P$ values for these statistical comparisons are shown in Appendix Table S1). Source data are available online for this figure.

observe cells with significantly higher foci numbers, again suggesting that other factors may contribute. To investigate this further, we tested whether the increase of Cas1-Cas2 foci is concentration-dependent. If the increase was entirely driven by ongoing normal DNA synthesis in the absence of cell division events, an increased saccharin concentration should show little effect. However, much in contrast to this prediction, we observed a significant exacerbation of foci numbers when switching from 1.4 to 2%, showing a dose-dependent effect (Fig. 2E,F). To explore the specificity of this phenotype, we tested another artificial sweetener, ace-K, which has previously also been reported to induce filamentation (de Dios et al, 2023). However, the ace-K treatment did not have a major impact on DNA replication dynamics, with the vast majority of cells having <8 foci, even at concentrations higher than those used for saccharin (Appendix Fig. S4). This confirms the active accumulation of a specific DNA substrate in saccharin-treated cells that is bound by Cas1-Cas2 complexes due to saccharin.

The data presented suggest that the lack of cell division likely will be the main contributor to the increase in the ploidy of cells upon saccharin exposure. However, the shift in the distribution to DNA synthesis foci numbers beyond those expected by a physiological duplication progression coupled to cell division made us hypothesise that repairing DNA damage may be responsible for the increased DNA synthesis observed in our microscopy and transcriptomic data, a point that is also supported by our observation that some foci amplification is observed in cells treated with saccharin in which use of the origin is not possible (Appendix Fig. S2).

## Does saccharin trigger DNA repair?

At this point, we have shown multiple independent lines of evidence (dRNA-seq, YPet-DnaN, *dnaA(ts)* and concentration-dependent effects, Dataset EV2, Fig. 2C–F; Appendix Fig. S2) to suggest that saccharin may be triggering DNA repair pathways. It was shown previously that saccharin induces DNA strand breaks in

mouse bone marrow cells that can be visualised by the COMET assay (Bandyopadhyay et al, 2008). Damaged DNA ends are the substrate for a process called break-induced replication (BIR), a mechanism where the invasion of a processed DNA end into a homologous template can trigger DNA synthesis away from the origin (Kockler et al, 2021). This repair pathway requires the processing of DNA intermediates by the replication restart pathways, PriA/PriB, PriA/PriC and PriC, to load the replicative helicase DnaB (Anand et al, 2013; Windgassen et al, 2018; Michel and Sandler, 2017). To distinguish between synthesis initiated at the *ori* and elsewhere within the chromosome to mediate repair after saccharin treatment, we quantified the number of replisomes per cell using the Cas1-Cas2 reporter in the wild-type *E. coli* or in mutants in *priB* or *priC*. If DNA synthesis triggered by saccharin treatment is mostly induced at the origin, and not by DNA damage, the *priB* and *priC* mutations should have little or no effect on the accumulation of Cas1-Cas2 foci. Indeed, treating with 1.4% saccharin produced an increase in Cas1-Cas2 foci in the wild-type as well in the Δ*priB* and Δ*priC* mutants (Fig. 3A,B), in line with synthesis being driven from *ori*. However, the accumulation of foci in both mutants was significantly lower than that observed for the wild-type *E. coli*. This would indicate that the accumulation of Cas1-Cas2 foci in response to saccharin exposure is partially dependent on the functional PriA/PriB and/or PriC restart pathways. If so, then DNA synthesis must have become blocked by some form of DNA damage, including, but not limited to, strand breaks followed by BIR. Importantly, the Δ*priB* and Δ*priC* mutants exhibited the same cell filamentation response as the wild-type upon saccharin exposure, therefore excluding the impact of inhibited cell division on overall foci counts. The impact of saccharin on DNA replication dynamics appears to be non-lethal however, as typically we did not observe cell lysis prior to 60 min (Fig. 1AB; Movies EV1 and EV2) but routinely observed impacts on DNA replication dynamics, confirming cell viability within this timeframe. This would suggest that bulge-mediated cell lysis is the primary mechanism through which saccharin mediates cell death. It would be intriguing to investigate the longer-term consequences

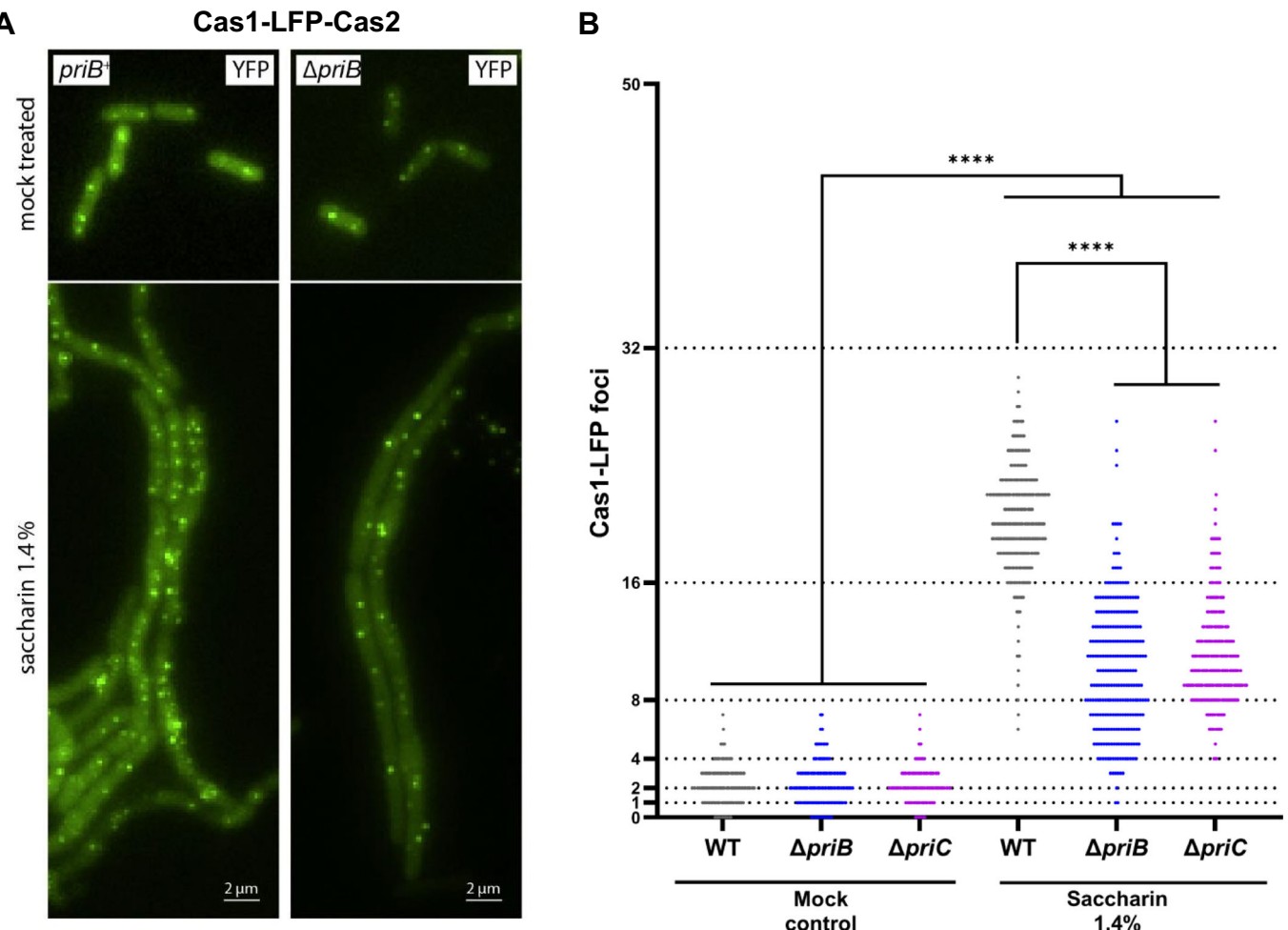

**Figure 3. Cas1-Cas2 foci formation in saccharin-treated cells is significantly reduced in cells lacking the replication restart proteins PriB and PriC.**

(A) Comparative visualisation of the effect of inactivating the replicative restart pathways (only ΔpriB is shown in this case, as the ΔpriC mutant showed a similar but slightly less pronounced phenotype). *E. coli* MG1655 cells with and without functional PriB or PriC restart proteins were grown to early exponential growth phase in LB broth (Miller) and either treated with saccharin as indicated, or mock-treated for 60 min. Simultaneously, expression of Cas1-linker-eYFP and Cas2 (called Cas1-Cas2 for simplicity) was induced by the addition of 0.1% arabinose. While in untreated cells, the absence of PriB or PriC did not result in a reduction in foci numbers, the filamented cells following saccharin treatment do show foci amplification in the absence of either PriB or PriC, but to a significantly lesser extent than in wild-type cells. The images shown are representative examples from a minimum of three biological replicates. The strains used were JD1708 (priB⁺ pTK135) and SP078 (ΔpriB pTK135). (B) Quantification of foci numbers from the experiments shown in (A). Dashed lines indicate the theoretically possible foci numbers expected from a normal duplication progression. The number of cells analysed, pooled from three biological replictaes, is 1313, 795 and 507 for wild-type, ΔpriB and ΔpriC, respectively, for the mock-treated samples, and 561, 274 and 297 for wild-type, ΔpriB and ΔpriC, respectively, for the saccharin-treated samples. Dashed lines indicate the theoretically possible foci numbers expected from a normal duplication progression. Results shown in (A) were analysed by Kruskal–Wallis test with Dunn's correction. Significance is indicated as ****$P \leq 0.0001$ (exact $P$ values for these statistical comparisons are shown in Appendix Table S1). Source data are available online for this figure.

of these impacts on DNA synthesis and replication, but due to the bulge-mediated lysis of the cells this is not possible within the current experimental setup.

## Saccharin inhibits the growth of clinically relevant pathogens

To this point, saccharin has shown an inhibitory activity on an *E. coli* lab model, directly impacting cell division and morphology, as well as DNA replication dynamics. Furthermore, our transcriptomics analysis showed that *E. coli* might be responding to saccharin by activating similar pathways to those triggered by β-lactam antibiotics. Indeed, our group and others previously

showed that saccharin can inhibit the growth of clinically relevant pathogens, such as *A. baumannii* and *P. aeruginosa* (Sünderhauf et al, 2020; Yu et al, 2022; Yu and Guo, 2022; de Dios et al, 2023). Altogether, the evidence for the potential of saccharin to inhibit bacterial growth by impacting different pathways prompted us to comprehensively test its antimicrobial effect on clinically relevant pathogenic isolates.

To uncover the full spectrum of saccharin antimicrobial activity, as well as the levels of inhibition, we applied a range of concentrations to a panel of high-priority pathogen clinical isolates, including MDR *E. coli*, *S. aureus*, *K. pneumoniae*, *A. baumannii* and *P. aeruginosa*. Saccharin was capable of inhibiting growth of all the strains tested in a dose-dependent manner (Fig. 4A–E). However, the relative levels of

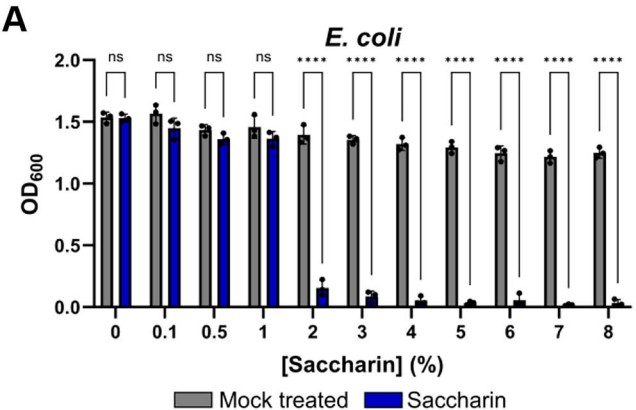

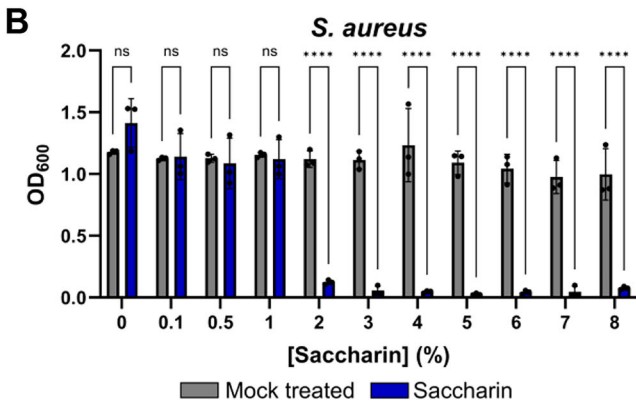

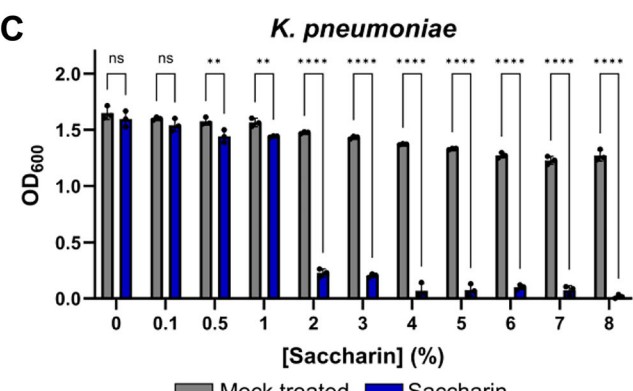

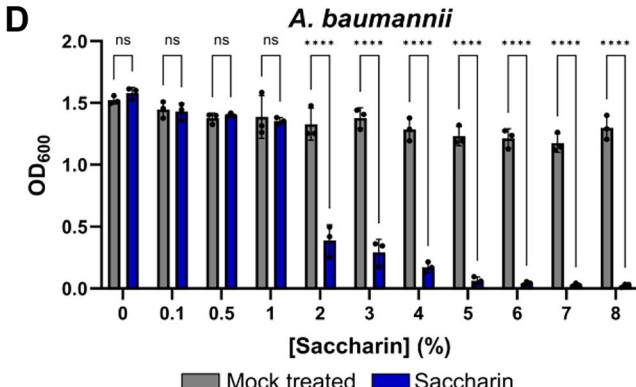

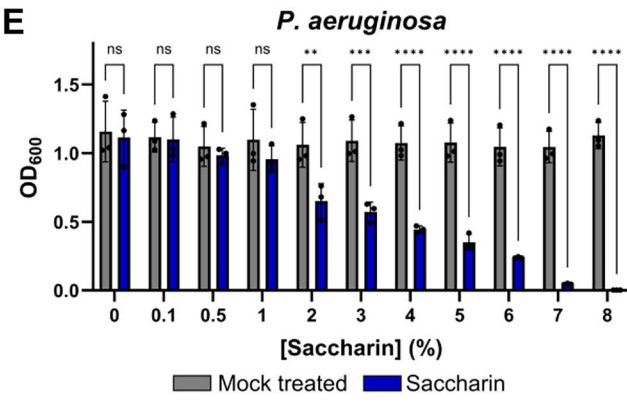

**Figure 4.   Saccharin displays dose-dependent growth inhibitory effects against both Gram-positive and Gram-negative pathogens.**

Growth inhibition of *E. coli* NCTC 13476 (**A**), *S. aureus* CCUG 68792 (**B**), *K. pneumoniae* ST234 (**C**), *A. baumannii* AB5075 (**D**) and *P. aeruginosa* PA14 (**E**) at a range of saccharin concentrations (0, 0.1, 0.5, 1, 2, 3, 4, 5, 6, 7, 8%). Inhibition was shown to be dose-dependent across all tested pathogens. Average values ± SD from three biological replicates are represented. Statistical analysis consisted of two-way ANOVA with Sidak's correction between the saccharin-treated samples and the water controls, for all panels. Significance is indicated as **$P \leq 0.01$, ***$P \leq 0.001$, ****$P \leq 0.0001$, ns = non-significant (exact $P$ values for these statistical comparisons are shown in Appendix Table S1). Source data are available online for this figure.

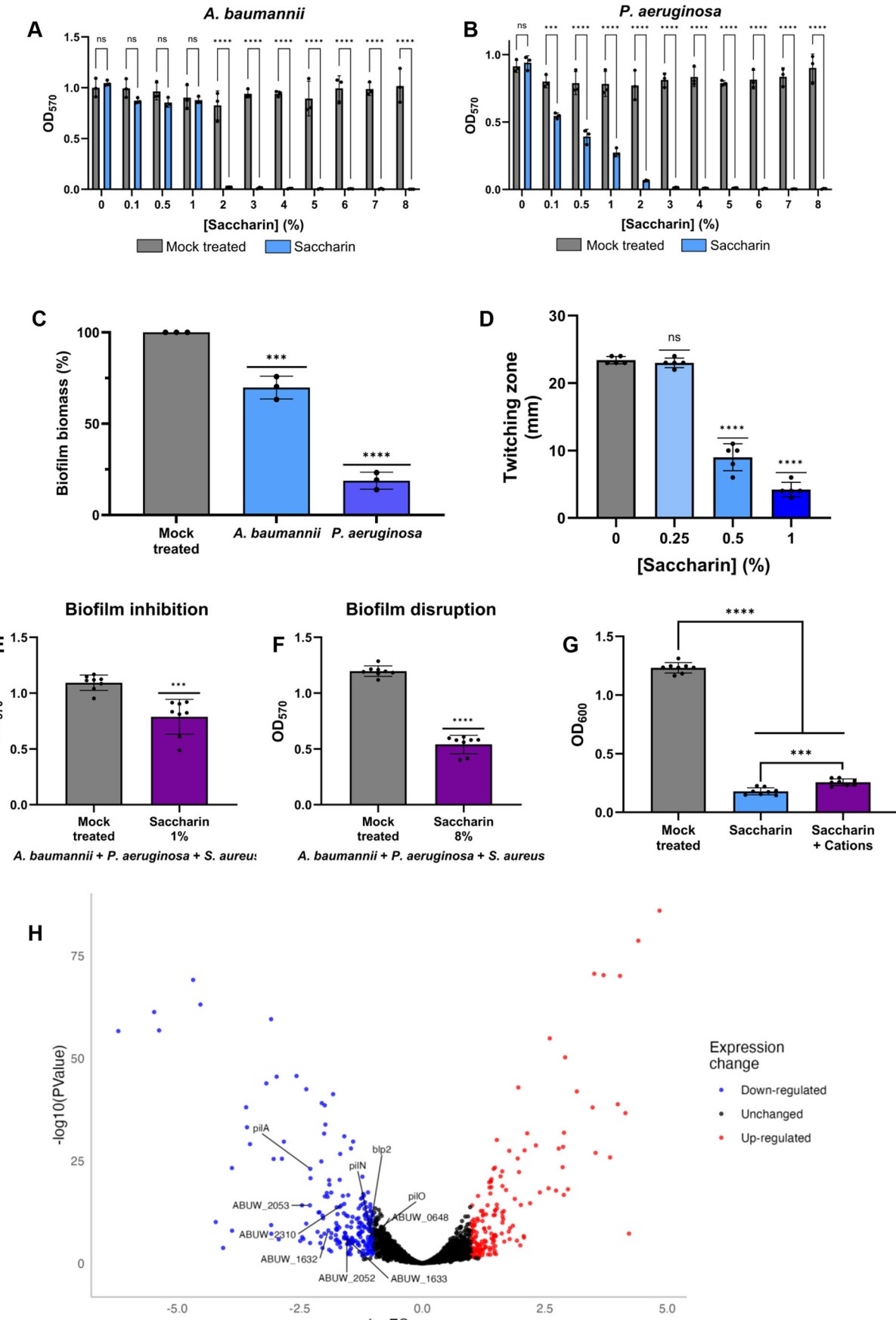

**Figure 5. Saccharin inhibits twitching motility and biofilm formation, and can disrupt preformed biofilms, even in a polymicrobial setting.**

Inhibition of de novo biofilm formation of *A. baumannii* (A) and *P. aeruginosa* (B) at a range of saccharin concentrations (0, 0.1, 0.5, 1, 2, 3, 4, 5, 6, 7, 8%). Inhibition showed to be dose-dependent across the range of concentrations. Average values ± SD from three biological replicates are represented. Statistical analysis for (A, B) consisted of two-way ANOVA with Sidak's correction between the saccharin-treated samples and the water controls. (C) The ability of an 8% saccharin solution to disrupt preformed *A. baumannii* and *P. aeruginosa* biofilms was also assessed. For both pathogens, the treatment produced a 30.2% and an 81.2% reduction in biofilm biomass, respectively. The percentage of biofilm biomass was calculated as the biofilm formation measurement for the treated samples with respect to the untreated control for each species. Average values ± SD from three biological replicates are represented. Statistical analysis for (C) consisted of one-way ANOVA with Dunnett's correction between the saccharin-treated and mock-treated samples. (D) Twitching motility assays using *A. baumannii* AB5075 over a range of concentrations of saccharin (0–1%), showing a dose-dependent negative effect of saccharin on this behaviour. Average values ± SD from five biological replicates are represented. Statistical analysis for (D) consisted of one-way ANOVA with Dunnett's correction. (E) Inhibition of de novo biofilm formation in mixed cultures containing a 1:1:1 ratio of *P. aeruginosa*, *A. baumannii* and *S. aureus* in LB media supplemented with either 1% saccharin or a vehicle control and incubated at 37 °C. The treatment produced a significant reduction in the polymicrobial biofilm formation. Average values ± SD from eight biological replicates are represented. Statistical analysis for (E) consisted of Student's *t* test between the saccharin-treated samples and controls. (F) Disruption of preformed polymicrobial biofilms from mixed cultures containing a 1:1:1 ratio of *P. aeruginosa*, *A. baumannii* and *S. aureus* in LB media after treating with 8% saccharin or a vehicle control for 24 h. Average values ± SD from eight biological replicates are represented. Statistical analysis for (F) consisted of Student's *t* test between the saccharin-treated samples and controls. (G) The addition of cations (2 mM CaCl$_2$ and 1 mM MgSO$_4$) can partially rescue the growth inhibitory effect of saccharin. Average values ± SD from eight biological replicates are represented. Statistical analysis for (G) was done by one-way ANOVA with Tukey's correction. (H) Volcano plot representing dRNA-seq results comparing cells treated with 1% saccharin to a mock treatment. According to the dRNA-seq results, 165 genes were upregulated (red) and 215 were downregulated (blue). In the interest of visualisation, ABUW_0607 was removed from the volcano plot (a plot version including this gene is shown in Appendix Fig. S9). Three biological replicates from each sample group were compared, and the statistical analysis was performed using DESeq's exact test for differences between two groups of negative-binomial counts. Significance for (A–G) is indicated as **$P \leq 0.01$, ***$P \leq 0.001$, ****$P \leq 0.0001$, ns = non-significant significant (exact *P* values for these statistical comparisons are shown in Appendix Table S1). Source data are available online for this figure.

this inhibition varied between pathogens. *E. coli*, *S. aureus*, *K. pneumoniae* and *A. baumannii* (Fig. 4A–D) showed a major reduction in growth (>70%) at a concentration of 2%, whereas *P. aeruginosa* (Fig. 4E) required a concentration of 6% to achieve similar levels of growth inhibition (77% reduction in growth). As for the different pathogens tested (except for *P. aeruginosa*) a sudden drop in viability occurred within the 1–2% saccharin interval, we calculated IC50 values (concentration producing a 50% of total inhibition) in that range (Appendix Fig. S1). As a result, the IC50 ranged from 1.2 to 1.5%, except in the case of *P. aeruginosa*, for which the IC50 was 2.5%. These results indicate that saccharin is effective at inhibiting the growth of both Gram-negative and Gram-positive pathogens. Furthermore, the varying levels of growth inhibition in a species-specific manner may explain the variable effects of saccharin reported previously on the microbiome (Suez et al, 2014). In addition, this would support the hypothesis that saccharin could alter the populational balance of the microbiome, rather than impacting the fitness of all the members of this ecosystem alike.

## Saccharin disrupts biofilm formation and motility

Biofilm formation during infection represents a major healthcare issue, as this bacterial lifestyle allows pathogens to remain recalcitrant to antibiotic treatments and the rigours of the host immune system (Kumar et al, 2017; Jamal et al, 2018; Roilides et al, 2015). To explore if saccharin has anti-biofilm properties, we assessed its ability to inhibit de novo biofilm formation in two particularly notorious biofilm-forming pathogens, *A. baumannii* and *P. aeruginosa* (Mulcahy et al, 2014; Maslova et al, 2021; Harding et al, 2018). Their levels of biofilm formation supported a better resolution to test a possible anti-biofilm activity of saccharin, compared to the negligible biofilm formation observed for *E. coli* (even without treatment) in the conditions tested (Appendix Fig. S5). We observed that saccharin could indeed inhibit this behaviour, showing a reduction of 97.5% for *A. baumannii* and 91.7% for *P. aeruginosa* at a 2% concentration (Fig. 5A,B). This greater reduction in biofilm formation than in bacterial growth suggests that this sweetener may have anti-biofilm properties. This is specifically supported by the anti-biofilm profile of saccharin on *P. aeruginosa*

(Fig. 5B), where the steadier decrease in viability over saccharin concentrations (Fig. 4E) allows a better distinction between anti-microbial and anti-biofilm effect.

Disrupting established biofilms is a key clinical challenge, particularly with respect to wound and indwelling device-associated infections (Maslova et al, 2021; Pelling et al, 2019). To evaluate if saccharin could disrupt a mature biofilm, we conducted biofilm disruption assays. As previously observed for biofilm inhibition, saccharin could disrupt mature biofilms of *A. baumannii* and *P. aeruginosa* to different levels depending on the species (Fig. 5C). In the case of *A. baumannii*, a treatment with 8% saccharin resulted in a reduction of a 30% in the biofilm biomass, whereas for *P. aeruginosa*, the treatment achieved an 81% reduction in the biofilm biomass compared to the vehicle control. Together, this positions saccharin as a potential anti-biofilm agent, not only to prevent biofilm formation, but to disrupt established biofilms such as those associated with chronic infections.

Twitching motility is an important virulence behaviour for many *A. baumannii* strains, including AB5075. It is mediated by type IV pili, which play a dual role in this type of motility as well as in adhesion (Ronish et al, 2019; Ellison et al, 2022). We previously demonstrated that sub-inhibitory concentrations of the artificial sweetener ace-K could abolish twitching motility in AB5075 (de Dios et al, 2023). For this reason, we tested if this effect was replicated by a range of saccharin concentrations. As a result, we obtained a dose-dependent inhibitory effect of saccharin on the twitching motility of AB5075 (Fig. 5D). Remarkably, this inhibition occurred at sub-inhibitory concentrations of saccharin (1% and below), showing the anti-virulence potential of this sweetener against an MDR strain of the critical priority pathogen *A. baumannii*.

## Saccharin can inhibit and disrupt polymicrobial biofilms

Within an infection context, polymicrobial communities of multiple pathogens are more frequently observed than single-species biofilms (Gabrilska and Rumbaugh, 2015; Anju et al, 2022; Kulshrestha and Gupta, 2022; Maslova et al, 2021). Within these

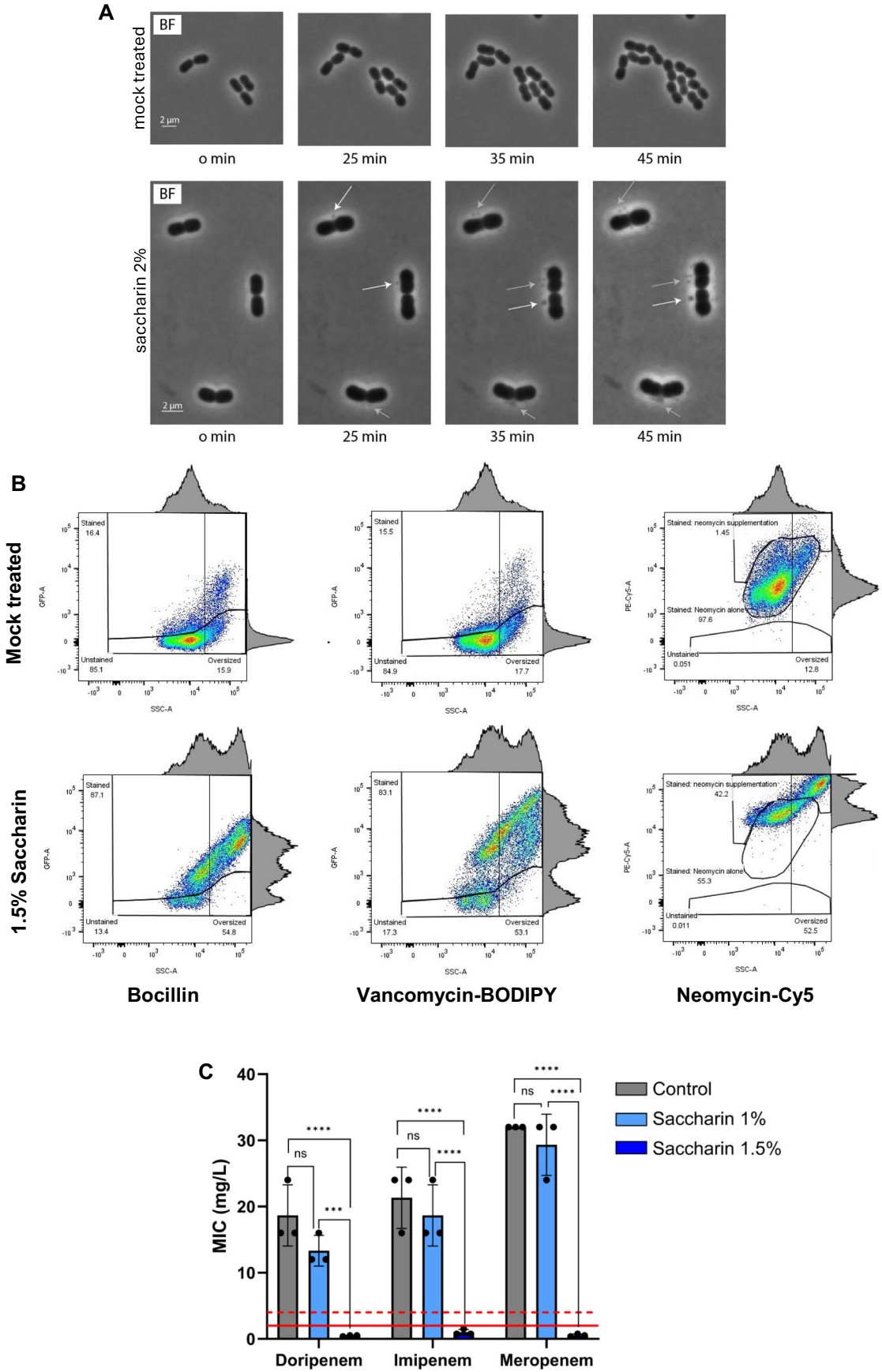

**Figure 6. Saccharin potentiates antibiotic activity against MDR *A. baumannii* AB5075 by increasing membrane permeability.**

(A) Impact of saccharin on cellular growth and morphology of *A. baumannii* observed during time-lapse phase contrast imaging. *A. baumannii* AB5075 cells grown on LB medium without sweetener (0%) were used as a vehicle control, highlighting the normal growth pattern of *A. baumannii*. In the presence of 2% saccharin, the images shown highlight the formation of membrane bulges (white arrows), which eventually rupture, resulting in the spilling of cytoplasmic content into the surrounding area (grey arrows). An environmental chamber was used to maintain a constant temperature of 37 °C. Cells were imaged for 60 min. A representative example out of three biological replicates is shown. (B) Exponential phase cells were treated with 1.5% saccharin or a mock treatment for 30 min. Then, they were incubated with 5 μM Bocillin, 5 μM Vancomycin- BODIPY™ FL or 0.1 μM Neomycin-Cy5 for 30 min prior to flow cytometry. The results show a saccharin dose-dependent increase in fluorescent antibiotic labelling (Fluorescence signal, *Y* axis) and cell size, which at this size range is best represented by the side scatter (SSC, *X* axis). Cell size is represented in the *X* axis (side scatter, SSCC) and fluorescence (fluorescent antibiotic labelling) in the *Y* axis. Cell populations were gated according to cell size and unlabelled controls as shown in Appendix Fig. S11 and explained in the Methods section. The images presented are a representative example of three biological replicates. A representative example of the full range of saccharin concentrations tested (0–1.5%), including all plots shown in (B), and average labelling and cell size trends obtained from three biological replicates are shown in Appendix Fig. S11. (C) We measured the MIC (E-test strips) for the carbapenems doripenem, imipenem and meropenem on Mueller-Hinton agar at sub-inhibitory concentrations of saccharin (1% and 1.5%) and a vehicle control. The results show a decrease of the MIC below the EUCAST breakpoint of each of the carbapenems at a concentration of 1.5% saccharin. The solid red line indicates the EUCAST breakpoint for doripenem and meropenem (2 mg/l); the dashed red line indicates the EUCAST breakpoint for imipenem (4 mg/l). In the control experiment, the imipenem MIC was above the detection limit of the E-test strip (32 mg/l). Average values ± SD from three biological replicates are represented. Statistical analysis for (C) was performed by two-way ANOVA with Tukey's correction within the sample groups indicated. ***$P = <0.001$, ****$P = <0.0001$, ns = non-significant (exact *P* values for these statistical comparisons are shown in Appendix Table S1). Source data are available online for this figure.

communities, cooperative interactions can take place, with different species protecting the whole community from insults, such as antibiotics, via the production of community resources. As a result, these polymicrobial biofilms remain much more recalcitrant to treatment and are commonly associated with chronic wounds (Kirketerp-Møller et al, 2020; Alvarado-Gomez et al, 2018; Maslova et al, 2021). As the three most common pathogens usually co-isolated from wounds are *P. aeruginosa, A. baumannii* and *S. aureus*, we tested the ability of saccharin to inhibit and disrupt a polymicrobial biofilm involving these three species (Dent et al, 2010; Dowd et al, 2008; Fazli et al, 2009). To confirm that a polymicrobial biofilm was formed, we grew a biofilm for 24 h starting with equal $OD_{600}$ amounts of each of the three species, disrupted it and cultured the resulting cell suspension in selective media for each pathogen. Despite expected variations in the species-to-species proportions by the end of the experiment, we could consistently recover all three pathogens from the biofilm, hence confirming its polymicrobial nature (Appendix Fig. S6). We then tested the effect of saccharin on the inhibition of de novo biofilm formation, observing that the presence of 1% saccharin resulted in a 28% reduction in biofilm formation (Fig. 6E). With respect to disruption of an established polymicrobial biofilm, the treatment with an 8% saccharin solution resulted in a 55% reduction of the biofilm biomass (Fig. 6F). The capacity to overcome this notoriously recalcitrant mode of multispecies growth highlights the therapeutic potential of saccharin.

## Saccharin produces pleiotropic gene expression changes in MDR *A. baumannii*

Despite having obtained conclusive mechanistic data showing the antimicrobial effect of saccharin using a lab *E. coli* strain, this model would not be representative of a clinically relevant scenario. For this reason, and given the urgent need for novel therapeutic approaches against MDR *A. baumannii*, we decided to conduct a dRNA-seq experiment using *A. baumannii* AB5075 as a model. We compared the presence of a sub-inhibitory concentration of saccharin (1%) to a mock treatment. This sub-inhibitory concentration was chosen to give a greater insight into the anti-virulence effects of saccharin without lethally impacting viability. In these

conditions, a total of 380 genes were more than 1-log differentially regulated, with 165 upregulated and 215 downregulated (Fig. 5H; Dataset EV3). To assess if there was any specific functional group of genes directly affected by the presence of saccharin, we performed a gene set enrichment analysis (GSEA) among the genes that were either up- or downregulated. As a result, we obtained that the only functional group significantly enriched among the downregulated genes (Dataset EV4) was that encompassing genes related to pili and fimbriae biogenesis as per their annotation (ABUW_0648, ABUW_1632, ABUW_1633, ABUW_2052, ABUW_2053 and ABUW_2310) (Gallagher et al, 2015). These genes are well known for their role in adhesion and biofilm formation in *A. baumannii* (Colquhoun and Rather, 2020). In addition to these, we could also find three genes belonging to the Type IV pili, including *pilN* and *pilO* (ABUW_0291 and ABUW_0292, respectively), which are part of the inner membrane complex, and the major pilin subunit *pilA* (ABUW_0304). This validates our previous findings showing the inhibitory effect of saccharin on the twitching motility of AB5075 (Fig. 5D). With respect to the upregulated genes, no functional enrichment was observed.

Other examples of gene groups that consistently appeared among the differentially regulated genes were either involved in iron homoeostasis or encoding iron-bound proteins (ABUW_1348, ABUW_1776, ABUW_2186, ABUW_2458, ABUW_2546, ABUW_2953, ABUW_3843, ABUW_3484, ABUW_3806) and multiple genes linked to sulfur metabolism (*cysN, cysT, cysW, sfnG, tauA, tauC, tauD*, ABUW_1021, ABUW_1941, ABUW_2169, ABUW_2335, ABUW_2336, ABUW_3853) according to their annotation (Gallagher et al, 2015). This suggested that iron and/or sulfate homoeostasis or acquisition pathways may be disturbed in the presence of saccharin. However, supplementing with neither ferric iron nor sulfate could alleviate the growth inhibition produced by saccharin (Appendix Figs. S7 and S8).

In addition to the previously mentioned gene expression alterations, a frequent signature we could observe among the differentially regulated genes was the association to the cell envelope. At least 67 differentially regulated genes were related to this structure (Dataset EV5) either directly as per their annotated gene description or being functionally associated to it (for example, being involved in transport, secretion or biofilm formation)

(Gallagher et al, 2015). This supported our previous data that membrane stability is compromised after saccharin treatment, thus producing an impact on fitness in combination with the disruption of DNA replication dynamics (Mitchell and Silhavy, 2019). As divalent cations are known to stabilise the cell envelope of Gram-negative bacteria by bridging the lipopolysaccharide molecules in the outer membrane, we further tested this hypothesis by supplementing the media with $Ca^{2+}$ and $Mg^{2+}$ (Clifton et al, 2015). This resulted in a partial relief of the growth inhibition (Fig. 5G). Although this rescue effect seemed modest with respect to the total inhibitory effect exerted by saccharin, this result is in line with the idea that this artificial sweetener disrupts the integrity of the cell envelope.

## Saccharin increases antibiotic transition across the cell envelope overcoming native resistance mechanisms

The damage to the membrane induced by saccharin exposure suggests that it could perturb the transition of molecules across the cell envelope, potentially impacting antibiotic uptake, and thus, susceptibility. To test this, we used MDR *A. baumannii* AB5075. Importantly for this pathogen, the low permeability of its cell envelope acts as an intrinsic antibiotic resistance mechanism (McCarthy et al, 2021). To first confirm that saccharin does induce an altered membrane morphology in *A. baumannii*, we used time-lapse microscopy on cells exposed to 2% saccharin. The treatment induced a loss of cell morphology with cells ballooning rather than filamenting as was observed with *E. coli* cells (Fig. 1A). Characteristic membrane bulges were also visible (Fig. 6A; Movies EV3 and EV4), suggesting that the membrane damaging effects of saccharin are consistent across species. Indeed, epifluorescence microscopy images of AB5075 cultures treated with a sub-inhibitory concentration of saccharin (1%) showed an increased permeability to DAPI than an untreated control (Appendix Fig. S10). This suggested that alterations in the *A. baumannii* envelope permeability induced by saccharin may increase the access of antibiotic to the cytosol. To further explore this, we tracked the uptake of multiple fluorescently labelled antibiotics after treating with a range of saccharin concentrations (0.5–1.5%) using a flow cytometry approach. The antibiotics selected were the penicillin fluorescent derivative Bocillin and a vancomycin-BODIPY™ FL conjugate (both penicillin and vancomycin inhibit peptidoglycan biogenesis by different mechanisms). We also wanted to assess if the transition of a non-cell envelope targeting antibiotic across the cell envelope was impacted by saccharin. To this end, we generated a bespoke neomycin probe (neomycin inhibits the ribosomal function) by conjugating it with Cy5, in analogy to a previously reported neomycin probe (Sabeti Azad et al, 2020). The various mechanisms of action of the selected fluorescent antibiotics would ensure that signal variations would depend on envelope permeability and not target availability or their physicochemical properties. Figure 6B shows the effect of 1.5% saccharin on cell morphology and antibiotic accumulation in *A. baumannii*, with both phenotypes being dose-dependent (Appendix Fig. S11). We could observe an increase in cell size, shifting from a 14.5% to 15.8% in oversized cell population with respect to total cells for all probes tested to 53.1%, 64.5% and 52.5% for Bocillin, Vancomycin-BODIPY™ FL and Neomycin-Cy5, respectively. These similar results suggest that changes in cell morphology are due to the saccharin treatment and

independent of the probe, and confirm the aberrant cell morphology induction by saccharin observed in our time-lapse experiments (Fig. 6A). On the other hand, we observed an increase in the populations labelled with the different probes, with shifts from 17.8 to 85.7%, 15.8 to 86.9% and 1.2 to 33.7% for Bocillin, Vancomycin-BODIPY™ FL and Neomycin-Cy5, respectively. This indicates that, although the permeability of the *A. baumannii* envelope was uneven for the different probes, all of them showed an increased cell penetration after saccharin treatment.

As a next step, we investigated if the increased cell envelope permeability of *A. baumannii* to antibiotics after saccharin treatment would lead to an increased sensitivity to antibiotics. Specifically, *A. baumannii* strains resistant to the last-resort antibiotics carbapenems, such as AB5075, have been listed at the top of the WHO priority pathogen list. For this, we assessed the minimum inhibitory concentration (MIC) of different carbapenems in the presence of 1% and 1.5% saccharin on cation-adjusted Mueller-Hinton agar. The treatment resulted in a dramatic dose-dependent decrease of the MIC, even dropping below the EUCAST breakpoint (2 mg/l for doripenem and meropenem, 4 mg/l for imipenem) at a 1.5% concentration of saccharin (Fig. 6C; Appendix Fig. S12). These findings support the potential of saccharin as an antibiotic potentiator, prospectively opening new avenues to tackle recalcitrant MDR infections.

## Saccharin-loaded hydrogels decrease bacterial burden in an ex vivo burn wound model

Given the emerging body of evidence that saccharin can inhibit bacterial growth and disrupt pathogenic behaviours such as biofilm formation, we next sought to explore its therapeutic potential. Wound healing is usually affected by the formation of biofilms, characteristic of chronic infection. Strikingly, there are relatively few topical antimicrobials in the drug development pipeline with anti-biofilm properties (WHO, 2022). In these situations, the preferred option is the application of the treatment by means of a hydrogel. This means of topical application outperforms others, such as soaked gauzes, due to the negative impact they may have on healing because of maceration and the lack of effective exudate management (Chamanga, 2015). For this reason, we formulated a saccharin-loaded tetraborate-PVA hydrogel. To firstly test its efficacy, we applied it for 1 h on early-stage *A. baumannii* AB5075 colony biofilms grown for 3.5 h on agar using 8% saccharin in the hydrogel compared to a vehicle control, a commercial silver-alginate wound dressing and an untreated control (Appendix Fig. S13). As this application showed promising results, we decided to challenge the efficacy of the saccharin hydrogel by applying it on a mature AB5075 biofilm, grown for 24 h, which is more recalcitrant to treatment. The application of the saccharin hydrogel to mature biofilms showed a dose-dependent reduction in the number of viable bacteria, even greater than that obtained with a commercial antimicrobial silver-alginate hydrogel, thus demonstrating the efficacy of this saccharin hydrogel formulation and its potential for clinical applications (Fig. 7A). After this test, we progressed to apply it as a wound dressing in an ex vivo burn wound model on porcine skin (de Dios et al, 2023). A single 1-h application of this 6% saccharin hydrogel formulation led to a 1.23-log reduction in bacterial numbers within the wound compared to the unloaded control hydrogel (Fig. 7B). These data

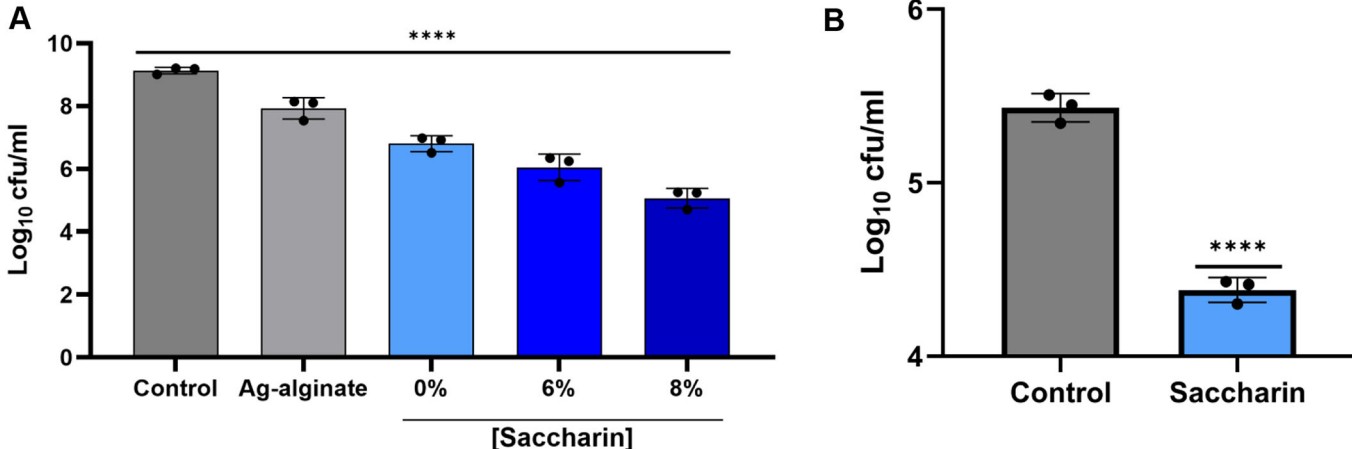

**Figure 7. Saccharin shows therapeutic potential in topical treatments.**

(A) Saccharin-loaded hydrogels (6% and 8%) produce a dose-dependent reduction in the bacterial numbers compared to a vehicle control when applied for 1 h on an *A. baumannii* AB5075 colony biofilm grown on an agar plate for 24 h. Average values ± SD from three biological replicates are represented. Statistical analysis for A was performed by one-way ANOVA with Tukey's correction. (B) A 6% saccharin-loaded hydrogel produces a significant reduction in bacterial numbers compared to a vehicle control in a porcine ex vivo burn wound model after a 1-h treatment. Average values ± SD from three biological replicates are represented. Statistical analysis for (B) was performed by Student's *t* test. ****$P = $ <0.0001 (exact *P* values for these statistical comparisons are shown in Appendix Table S1). Source data are available online for this figure.

demonstrate that saccharin retains its antimicrobial activity when loaded in a hydrogel.

## Discussion

Saccharin has been used as a sugar substitute in the human diet for over a century. However, its safety for human health has remained a matter of controversy. In spite of this, it is used as an artificial sweetener worldwide, being the most consumed sweetener in relation to its sweetening power (Sylvetsky and Rother, 2016). In recent years, an increasing number of studies have linked saccharin to different deleterious effects for human health, including glucose intolerance and inflammatory processes. Interestingly, a common nexus between saccharin and these effects is the gut microbiome, pointing at a direct effect of this sweetener on bacteria (Suez et al, 2014; Bian et al, 2017; Skurk et al, 2023; Del Pozo et al, 2022). Nevertheless, despite the unequivocal evidence of the impact of saccharin on the microbiome, its direct effect on the bacterial physiology and its underlying mechanism remains unexplored.

### Impact on bacterial DNA replication dynamics

Saccharin exposure triggered a loss of cell morphology and filamentation in an *E. coli* model. Z-ring formation did not appear to be impacted by saccharin, indicating an alternative trigger for cell filamentation (Fig. 1B). When we explored the consequences of saccharin exposure at the transcriptomic level, several genes associated with the cell envelope were differentially regulated, but strikingly, pathways linked to DNA replication and mismatch repair systems appeared significantly upregulated (Dataset EV2). This suggested saccharin could be impacting DNA synthesis dynamics within the cell. We confirmed this using several different

DNA replication-associated fluorescent reporters, each showing saccharin exposure triggered increased signal. The majority of these increases can be attributed to chromosomal replication continuing unabated in filamented cells. However, we consistently observed a subset of cells in which the signal accumulated to higher than anticipated levels, suggesting additional effects. We demonstrated the specificity of this effect by revealing it was dose-dependent and specific to saccharin as ace-K, another artificial sweetener which can induce filamentation, did not trigger the same pronounced effect. Given the increase in non-*ori*-initiated DNA synthesis activity following saccharin treatment (Fig. 2C–F), we hypothesised that saccharin might be causing DNA damage. This would be coherent with previous studies showing saccharin-mediated DNA damage via the COMET assay in mouse bone marrow cells (Bandyopadhyay et al, 2008), saccharin-induced sperm DNA fragmentation (Rahimipour et al, 2014) and that this sweetener triggers the SOS DNA repair system and the production of reactive oxygen species in bacteria (Yu et al, 2021; Yu and Guo, 2022). In bacteria, BIR pathways catalyse the initiation of DNA synthesis away from the *ori* to repair DNA lesions, therefore we measured the replication foci numbers in two mutants defective in the BIR pathways (*priB* and *priC*) using the recently developed Cas1-Cas2 DNA synthesis reporter (Fig. 3A,B). This reporter showed significantly lower foci numbers in the absence of the restart protein PriB and PriC upon saccharin exposure, indicating that saccharin induces DNA synthesis away from the *ori* region. The involvement of the replication restart pathways indicates that ongoing synthesis must be blocked at obstacles, at least in some cells. Based on the mouse bone marrow data a certain degree of fragmentation and break-induced replication certainly would be in line with such a result. However, other repair pathways might also explain the data, in line with our transcriptomics data in *E. coli*, or indeed both BIR and additional repair pathways might be triggered.

Our attempts to establish the molecular details were thwarted by the fact that all our assays to study DNA synthesis dynamics were time-limited as a result of cells lysing. Thus, the impact of saccharin on the cell envelope usurps the physiological consequences of altered DNA synthesis dynamics within the current experimental setup. It will require future work to identify what specifically is happening to the bacterial chromosome to trigger this increase in DNA synthesis and uncovering the multi-generational consequences of saccharin exposure on bacterial genome integrity. However, clearly some level of DNA damage is caused by this artificial sweetener. While non-lethal in our experimental conditions, the impact on DNA replication dynamics is likely to ultimately compromise genome integrity, leading to an accumulation of mutations in the chromosome.

## Impact on the cell envelope

Throughout this work, we have shown different results pointing towards an impact of saccharin on the cell envelope that may explain the filamentation phenotype. Our microscopy experiments on *E. coli* have shown that cells filament and eventually lyse due to membrane bulging (Fig. 1; Movie EV2). This is a typical signature of β-lactam antibiotics, which target peptidoglycan biosynthesis (Yao et al, 2012). Intriguingly, our transcriptomics dataset comparing *E. coli* cells treated and untreated with saccharin showed activation of β-lactam resistance pathways (Dataset EV2). In *A. baumannii*, we similarly observe a loss in cell morphology, resulting in non-dividing, enlarged spherical cells and characteristic cell bulging after saccharin treatment (Fig. 6A; Movie EV4). This behaviour is strikingly similar to that shown by Dorr et al after treating *A. baumannii* with meropenem (Dörr et al, 2015). Further studies by Bailey et al have also shown that mutations disrupting genes involved in peptidoglycan biosynthesis in *A. baumannii* produce "giant cells" that are strongly sensitised to meropenem (Bailey et al, 2019; Bailey et al, 2023). Interestingly, mutation of genes involved in DNA replication impaired cell division and also sensitised to this β-lactam antibiotic (Bailey et al, 2023). This aligns with our findings showing the impact of saccharin on the cell envelope morphology and permeability and its potentiating effect on carbapenems (Fig. 6). There were no genes directly associated with envelope stress response pathways differentially regulated in the dRNA-seq analysis performed with *A. baumannii*, which may suggest that the affected envelope-related genes respond to a general remodelling in the global transcription to mitigate the saccharin impact. However, it is worth noting that this experiment was conducted using 1% saccharin specifically to explore the sub-lethal anti-virulence effects of saccharin on *A. baumannii*. In addition, we cannot discard envelope stress responses being activated at post-transcriptional levels, which would not be disclosed by transcriptomics. Interestingly, the analysis of our *A. baumannii* transcriptomics data did not show the same enrichment pattern. This could be due to species-to-species differences in the saccharin sensitivity, whereby DNA replication effects are masked in *A. baumannii* or occur earlier upon exposure. Although further efforts need to be made to fully address its mechanism of action, the conserved impact on the cell envelope of different species reinforces our hypothesis of the β-lactam-like effect of saccharin and opens a new avenue to use as the basis for novel therapeutic approaches.

## Therapeutic potential of saccharin

Currently, the alarmingly increasing prevalence of antimicrobial-resistant infections worldwide has led to one of the most important concerns of our time: the antimicrobial resistance crisis. For this reason, novel therapeutic approaches need to be urgently developed against pathogenic bacteria that have evolved resistance to most of the frontline and last-resort antibiotics. We show that saccharin has potential as a potent antimicrobial and can significantly inhibit the growth of MDR *E. coli*, *S. aureus*, *K. pneumoniae*, *A. baumannii* and *P. aeruginosa*. (Fig. 4). We further show that saccharin can inhibit biofilm formation (Fig. 5A,B), which is a cause of infection treatment failure. However, a more frequent situation in the clinical setting is that a biofilm has already been formed when an antimicrobial treatment starts. For this reason, we challenged saccharin to disrupt preformed biofilms of both *A. baumannii* and *P. aeruginosa*, obtaining significant reductions in the biofilm biomass in both cases (Fig. 5C). These two species, as well as *S. aureus*, are the most prevalent pathogens isolated from wound infections, an environment in which the most common scenario is the polymicrobial infection (Alvarado-Gomez et al, 2018). To further challenge the efficacy of saccharin in disrupting biofilms, we reproduced these assays on polymicrobial biofilms, obtaining significant reductions in the de novo biofilm formation and biofilm biomass of preformed biofilms (Fig. 5E,F). It is estimated that biofilms produce a global burden of $386.8 billion in the healthcare systems and are associated to a higher rate of treatment failure (Cámara et al, 2022). In this scenario, the anti-biofilm properties of saccharin can be harnessed to develop new therapeutic strategies that help to increase the efficiency of antimicrobial treatments, increasing the success rate and reducing the costs.

Most antibiotics need to find their way into the cell to produce their antimicrobial effect, crossing the bacterial cell envelope by different uptake mechanisms. This can be a limiting factor in the efficacy of an antibiotic. For example, the membrane impermeability to antibiotics is one of the intrinsic resistance mechanisms of *A. baumannii* (McCarthy et al, 2021). Thus, disrupting this barrier can lead to more efficient antimicrobial therapies and potentiate the activity of commonly used antibiotics. In this regard, as saccharin produced an alteration of the cell envelope stability (Figs. 1 and 6A), we investigated if this could affect the function of the membrane as a barrier. We were able to validate an increased permeability of MDR *A. baumannii* AB5075 cells at sub-inhibitory concentrations of saccharin, which led to increased uptake of the fluorescently labelled antibiotics (Fig. 6B). To challenge if this would lead to an increased antibiotic sensitivity, we assessed the minimum inhibitory concentration (MIC) of different carbapenems on *A. baumannii* AB5075, which is completely resistant to these last-resort antibiotics (Jacobs et al, 2014). The reduction of the MIC to levels below the EUCAST breakpoint for *A. baumannii* to be considered resistant (Fig. 6C) highlights the potential of saccharin to be used in combination therapies to increase antibiotic uptake and tackle antibiotic-resistant infections.

As a final step, we tested the therapeutic potential of saccharin to be used as an antimicrobial. As the concentrations of saccharin with antimicrobial activity can be challenging to implement therapeutically when administered orally or intravenously, we opted to propose saccharin as a topical antimicrobial. To assess the feasibility of saccharin as a topical treatment, we selected an ex vivo model of

clinically relevant conditions, such as a burn wound (Fig. 7B), applying a PVA hydrogel augmented with saccharin. Hydrogels allow a controlled and prolonged release of treatment on wounds (Firlar et al, 2022). However, maintaining the activity of the active compound in the hydrogel for the duration of the treatment can be challenging. In the case of saccharin, we were able to formulate a stable hydrogel that retained the antimicrobial activity. After testing it initially in vitro, we successfully applied it on an ex vivo burn wound model using porcine skin and simulating a biofilm-associated *A. baumannii* infection (de Dios et al, 2023). Together, these pilot ex vivo topical applications demonstrate the feasibility of saccharin-based formulations as an inexpensive option for prophylaxis and treatment against wound infections. Future work will explore these formulations in acute and chronic in vivo infection models.

As the realities of a potential post-antibiotic era are beginning to be seen with the increasing prevalence of pan-drug-resistant pathogens in hospitals, the urgent need for novel therapeutics is being exacerbated. Here, we have identified a compound in saccharin with previously unrecognised therapeutic potential, overcoming classical limitations associated with many frontline antibiotics, such as the inability to treat established biofilms or to be integrated into hydrogel wound dressings. It also has the capacity to potentiate the activity of frontline antibiotics, re-sensitising previously resistant bacteria. The future therapeutic development of non-classical antimicrobials such as saccharin is likely to be critical to our capacity to effectively control and treat MDR pathogens in the coming decades.

# Methods

### Reagents and tools table

| Reagent/resource | Reference or source | Identifier or catalogue number |
|---|---|---|
| **Experimental models** | | |
| *Klebsiella pneumoniae* ST234 | Nordmann et al, 2012 | |
| *Pseudomonas aeruginosa* PA14 | Schroth et al, 2018 | |
| *Staphylococcus aureus* CCUG 68792 | CCUG | |
| *Escherichia coli* NCTC 13476 | NCTC | |
| *Acinetobacter baumannii* AB5075 | Jacobs et al, 2014 | |
| *E. coli* K12 MG1655 | Bachmann, 1996 | |
| *E. coli* K12 AB1157 RRL189 | Reyes-Lamothe et al, 2008 | |
| *E. coli* K12 MG1655 AS1062 | This work | |
| *E. coli* K12 MG1655 JD1459 | This work | |
| *E. coli* K12 MG1655 JD1708 | Killelea et al, 2023 | |
| *E. coli* K12 MG1655 JD1709 | de Dios et al, 2023 | |
| *E. coli* K12 MG1655 JD1716 | Killelea et al, 2023 | |
| *E. coli* K12 MG1655 SP078 | This work | |

| Reagent/resource | Reference or source | Identifier or catalogue number |
|---|---|---|
| *E. coli* K12 MG1655 SP079 | This work | |
| **Recombinant DNA** | | |
| pTK135 | Killelea et al, 2023 | |
| pTK136 | Killelea et al, 2023 | |
| **Antibodies** | | |
| **Oligonucleotides and other sequence-based reagents** | | |
| **Chemicals, enzymes and other reagents** | | |
| Sodium saccharin | Fisher Scientific | 10722271 |
| LB broth | Fisher Scientific | 244620 |
| Mueller-Hinton broth | Merck | 70192-500G |
| $CaCl_2$ | Fisher Scientific | 10403265 |
| $MgSO_4$ | Fisher Scientific | 10256840 |
| Tryptone | Fisher Scientific | 211705 |
| Yeast extract | Fisher Scientific | H26769.36 |
| Agar | Merck | 05039-500 G |
| Arabinose | Fisher Scientific | 10509980 |
| Crystal violet | Fisher Scientific | 11445027 |
| Ethanol 99% | Fisher Scientific | 10048291 |
| RNAlater | Fisher Scientific | 10391085 |
| 10-nonyl acridine orange | Invitrogen | A1372 |
| SeaKem LE Agarose | Lonza | 50002 |
| M9 minimal medium 5X | Sigma-Aldrich | M9956-500ML |
| Bocillin-FL penicllin | ThermoFisher Scientific | B13233 |
| Vancomycin-BODIPY | ThermoFisher Scientific | V34850 |
| Neomycin-Cy5 | This work | |
| Doripenem E-test | bioMérieux | 432126 |
| Imipenem E-test | bioMérieux | 412374 |
| Meropenem E-test | bioMérieux | 412402 |
| Polyvinyl alcohol | Merck | 363138-500G |
| Potassium tetraborate | Merck | P5754-500G |
| Molecular sieves, 4 Å | VWR | 28464.295 |
| Thin-layer chromatography plates | Supelco | 1.05570.0001 |
| $KMnO_4$ | Fluka | 60470 |
| Preparative thin-layer chromatography plates | Merck | 1.05744.0001 |

| Reagent/resource | Reference or source | Identifier or catalogue number |
|---|---|---|
| CDCl$_3$ | VWR | 87153.0100 |
| Neomycin trisulfate salt hydrate | Sigma-Aldrich | N1876-25G |
| N,N-Dimethylformamide (DMF) | VWR | 23466.298 |
| Triethylamine | Sigma-Aldrich | 8.08352.1000 |
| Di-tert-butyl dicarbonate | Sigma-Aldrich | 8.18282.0100 |
| Ethyl acetate | VWR | 23882.321 |
| NaCl | VWR | 27800.291 |
| Na$_2$SO$_4$ | Supelco | 1.06649.1000 |
| Silica gel | VWR | 84894.360 |
| Methanol | VWR | 20846.326 |
| Dichloromethane | VWR | 23367.321 |
| Pyridine | Merck | 1.09728.1000 |
| 2,4,6-Triisopropylbenzenesulfonyl chloride | TCI | T0459 |
| 2-Aminoethanethiol hydrochloride | TCI | A0296 |
| Toluene | Supelco | 1.08325.1000 |
| Trifluoroacetic anhydride | Fluka | 91720 |
| NaHCO$_3$ | Merck | 1.06329.1000 |
| Diethyl ether | VWR | 23811.292 |
| Pentane | VWR | 26185.297 |
| Sodium ethoxide (2.8 M solution in ethanol) | Sigma-Aldrich | 230553-100 ML |
| Ethanol (absolute, for synthesis) | Kemetyl Norge AS | |
| K$_2$HPO$_4$ | Fluka | 60230 |
| NH$_4$OH (28% solution in water) | Fluka | 09858-1L |
| Cyanine5 monofunctional NHS-ester | Lumiprobe | 23020 |
| N,N-Diisopropylethylamine | Merck | 8.00894.0100 |
| Chloroform | VWR | 22706.292 |
| Trifluoroacetic acid | Thermo Scientific | 139721000 |
| Hydrochloric acid | VWR | 20248.364 |
| **Software** | | |
| DESeq | Love et al, 2014 | |
| bcl2fastq | Illumina | 4.2.4 |
| HISAT | Kim et al, 2019 | 2.2.0 |
| limma | Ritchie et al, 2015 | |
| FUNAGE-Pro | de Jong et al, 2022 | 2.0 |
| NIS Elements-BR | Nikon | 4.51 |
| Adobe Photoshop CC | Adobe | 24.0.0 |
| BD FACSDiva | BD Biosciences | 9.0 |
| FlowJo | BD FlowJo | 10.8.1_CL |
| GraphPad Prism | GraphPad Software | 10.1.2 (324) |

| Reagent/resource | Reference or source | Identifier or catalogue number |
|---|---|---|
| ChemDraw Professional | Revvity Signals | 23.1.1.3 |
| MestReNova | Mestrelab Research | 14.2.1 |
| Xcalibur | Thermo Scientific | |
| MassHunter | Agilent | |
| **Other** | | |
| Bioanalyzer Instrument | Agilent Technologies | 2100 |
| MiSeq | Illumina | MiSeq |
| RNAeasy purification kit | Qiagen | 74106 |
| Bioanalyzer RNA 6000 Nano | Agilent Technologies | 5067-1511 |
| GeneFrame | Thermo Scientific | AB0577 |
| Ti-U Inverted microscope | Nikon | |
| CFI Plan Fluor DLL 100 × objective | Nikon | |
| ORCA Flash 4.0 LT plus camera | Hamamatsu | |
| pE-100 single LED wavelength source | CoolLED | |
| pE-4000 illumination system | CoolLED | |
| Environmental chamber | Digital Pixel | |
| BD FACSAria III Cell Sorter | BD Biosciences | 2015-07-14 T |
| Custom 3D-printed hydrogel mould | This work | |
| Mars 2 Pro 2 K resin 3D printer | Elegoo | |
| Wash & Cure 2.0 chamber | Anycubic | |
| Pig skin | Fine Food Specialists Ltd. | |
| Puriflash XS420 with integrated UV/ELS detector | Interchim | |
| Avance III HD NMR spectrometer, 9.4 Tesla, with SmartProbe | Bruker | |
| LTQ Orbitrap XL mass spectrometer with ESI ionisation | Thermo Scientific | |
| 6540B Q-ToF mass spectrometer with ESI ionisation source, coupled with 1290 Infinity UHPLC system | Agilent | |

## Bacterial strains and growth conditions

*A. baumannii* AB5075 (VIR-O phase variant (Chin et al, 2018)), *P. aeruginosa* PA14, *E. coli* NCTC 13476 (and *E. coli* K12 derivatives), *K. pneumoniae* ST234 and *S. aureus* CCUG 68792 were routinely grown in LB (Miller) broth or agar at 37 °C (180 rpm shaking for broth cultures), unless otherwise stated. When necessary, LB was supplemented with sodium saccharin (at the concentration specified in each experiment), CaCl$_2$ 2 mM, MgSO$_4$ 1 mM, 100 μM FeCl$_3$, 300 μM FeCl$_3$ and/or 5 mM K$_2$SO$_4$. All strains used in this work are listed in Dataset EV6.

## E. coli K12 derivative strains: construction and expression induction

E. coli K12 AB1157 and MG1655 derivative strains used for microscopy experiments are described in the Dataset EV6. Strains were constructed via P1vir transductions (Thomason et al, 2007) or by single-step gene disruptions (Datsenko and Wanner, 2000), as indicated in Dataset EV6. For the expression of Cas1 fused via a linker to eYFP (named Cas1-Cas2 for simplicity) and Cas2, we used plasmid pTK135 (Killelea et al, 2023). Briefly, pBAD-HisA (Invitrogen) was used for the expression of Cas1-LFP-Cas2 under control of the arabinose-inducible $P_{araBAD}$ promoter, resulting in the over-expression of both Cas1-LFP and untagged Cas2, allowing formation of functional Cas1-Cas2 complexes that can be visualised in living cells (Killelea et al, 2023). For expression of Cas1[R84G]-LFP, we used pTK136, which is as pTK135, but with a mutated Cas1 version, as previously described (Killelea and Dimude et al, 2023). For the experiments using pTK135 and pTK136, expression was induced by addition of arabinose to a final concentration of 0.1% for 60 min before the cells were imaged, as described below. pCP8 is a multicopy pBR322 derivative carrying p$ftsKi$-$ftsZ$-$cfp$. The plasmid allows the constitutive low expression of a FtsZ-CFP fusion protein from a weak promoter within FtsK (Wang et al, 2005).

### Growth and biofilm inhibition assays

Overnight cultures of A. baumannii AB5075, P. aeruginosa PA14, E. coli NCTC 13476, K. pneumoniae ST234 and S. aureus CCUG 68792 were diluted in 96-well plates to $OD_{600}$ 0.1 in LB medium supplemented with saccharin in concentrations ranging from 0.5 to 8% (weight/volume expressed in grams per 100 ml), including a vehicle control for each dilution. Cultures were incubated at 37 °C, 180 rpm for 18 h. Following incubation, planktonic growth was assessed by $OD_{600}$ reading.

In the case of A. baumannii AB5075 and P. aeruginosa PA14 cultures biofilm formation was measured after measuring planktonic growth. Media and planktonic cells were removed from the wells, and biofilms were gently washed with deionised water three times. Then, 200 μL of 0.1% crystal violet was added to each well and plates were incubated statically for 10 min at room temperature. The stain was then removed, and wells were washed five times with deionised water. After leaving the plates to air-dry, the retained crystal violet was re-solubilised by adding 200 μL of 99% ethanol to each well and incubating statically at room temperature for 6 h. Crystal violet was quantified by measuring absorbance at 570 nm.

### Biofilm disruption assay

To assess the ability of saccharin to disrupt established biofilms, overnight A. baumannii AB5075 and P. aeruginosa PA14 cultures were diluted in 96-well plates to $OD_{600}$ 0.1 in LB medium. Plates were incubated for 18 h at 37 °C, 180 rpm to allow biofilms to form. Following incubation, the growth medium was removed from the wells and biofilms were washed three times with 200 μL of sterile PBS to remove any unbound planktonic cells. Fresh LB medium supplemented with 8% saccharin or vehicle control was added to the wells. Plates were incubated for a further 24 h at 37 °C, 180 rpm.

Following this treatment, biofilms were stained with 0.1% crystal violet as detailed above. The reduction in biofilm was represented as a percentage reduction compared to the control.

### Polymicrobial biofilm dispersal

To assess the effect of saccharin on biofilm formation by polymicrobial cultures and the ability of saccharin to disrupt established polymicrobial biofilms, biofilm inhibition and biofilm disruption assays were set up as previously detailed with some amendments. For the polymicrobial MBIC experiments cultures of A. baumannii AB5075, P. aeruginosa PA14 and S. aureus CCUG 68792 were co-inoculated in a 1:1:1 ratio into TSB containing either 2% or 4% saccharin in a 96-well plate. Plates were then incubated at 37 °C, 180 rpm for 24 h. Following incubation, cultures were removed, and biofilm was stained and quantified with crystal violet as previously described.

For biofilm disruption assays, mixed cultures of A. baumannii AB5075, P. aeruginosa PA14 and S. aureus CCUG 68792 (in a 1:1:1 ratio) were co-inoculated into TSB in a 96-well plate and incubated for 24 h at 37 °C, 180 rpm. Following incubation, media was removed from the plates and replaced with fresh LB containing 8% saccharin and incubated for a further 24 h at 37 °C, 180 rpm. Following this second incubation, biofilms were stained and quantified with crystal violet, as previously detailed.

### RNA sequencing and differential gene expression analyses

To assess transcriptomic changes in E. coli after saccharin treatment, we used similar conditions to those used in the microscopy experiments. Briefly, E. coli MG1655 overnight cultures were diluted 1/100 (v/v) in 20 ml of fresh LB broth (in biological triplicate per condition) and incubated at 37 °C, 180 rpm until early exponential phase ($OD_{600}$ 0.3). At this point, cultures were treated with a final concentration of 1.4% saccharin or an equivalent volume of water and resumed incubation (37 °C, 180 rpm) for an additional hour. Then, samples were withdrawn and preserved in RNAlater at −80 °C until further processing.

In the case of A. baumannii AB5075, overnight cultures were diluted 1/100 (v/v) in 20 ml fresh LB broth supplemented with either 1% saccharin or the matching volume of vehicle control (three biological replicates per condition). The cultures were grown to mid-exponential phase ($OD_{600}$ 0.6 approx.) at 37 °C, 180 rpm. Then, samples were withdrawn, spun down and preserved in RNAlater at −80 °C until further processing.

The total RNA from each sample was purified using the RNAeasy Kit with on-column DNAase digestion (Qiagen). RNA integrity was assessed using a Bioanalyzer (RNA 6000 Nano kit) according to the amplitude and sharpness of the peaks corresponding to the 16S and 23S rRNAs. Samples were further processed for cDNA library preparation and sequencing on an Illumina MiSeq with 12 million reads per sample. Sequencing and downstream bioinformatic analyses were performed at SeqCenter (Pittsburgh, Pennsylvania, USA). Quality control and adaptor trimming was performed with bcl2fastq (Version v4.2.4). Read mapping were performed with HISAT 2.2.0. Differential expression analysis was performed using the DESeq R package and using the E. coli MG1655 genome annotation (GenBank accession number

U00096.3; Blattner et al, 1997) or *A. baumannii* AB5075-UW genome annotation (GenBank accession number NZ_CP008706.1; Gallagher et al, 2015) as references. To assess the physiological pathways altered by saccharin in *E. coli*, a KEGG pathway analysis was performed using limma's "kegga" functionality (default parameters) (Ritchie et al, 2015), where a significant up- and downregulation was considered with an FDR < 0.05. For *A. baumannii*, the functional enrichment of significantly up- and downregulated gene subsets was assessed by means of a Gene Set Enrichment Analysis using FUNAGE-Pro (v2.0) with default parameters (de Jong et al, 2022).

### Twitching assay

The twitching motility of *A. baumannii* AB5075 was assessed on twitching agar (tryptone 10 g/l, yeast extract 5 g/l, NaCl 2.5 g/l, agar 10 g/l) as previously described (de Dios et al, 2023). Briefly, freshly autoclaved twitching agar was supplemented with 1%, 0.5% or 0.25% saccharin, or a vehicle control. 10 ml of the supplemented agar were poured in 90-mm Petri dishes and left to dry for 10 min next to a Bunsen burner. The twitching plates were stab-inoculated with a pipette tip with independent AB5075 colonies freshly grown on LB agar and incubated at 37 °C for 48 h.

### Single-image microscopy

Fresh overnight cultures of the *E. coli* strains of interest were diluted 100-fold in fresh LB broth (Miller composition) and incubated with vigorous aeration at 37 °C until $OD_{600}$ reached 0.2. If cells were grown in the presence of plasmids (see section "*E. coli* K12 derivative strains: construction and expression induction"), ampicillin was added to the culture at a final concentration of 50 µg/ml. Upon reaching $OD_{600}$ 0.2, 2 ml samples were transferred into sterile pre-warmed glass tubes, and saccharin was added to each tube in the desired concentration. The cells were then incubated for different times as indicated in each figure. For staining of the membrane, 10-nonyl acridine orange (NAO, Invitrogen) was added to the sample to a final concentration of 200 nM and incubated for 5 min at room temperature. The visualisation of origin and terminus areas was achieved by using strain RRL189 (see Dataset EV6), which constitutively expresses mCherry-LacI and eCFP-TetR, which in turn bind *lacO240* and *tetO240* operator arrays located in the origin and terminus area of the chromosome, respectively. 1 µl of the sample was then pipetted onto an agarose pad and air-dried. For generation of pads, a 65 µl (15 × 16 mm) GeneFrame (Thermo Scientific) was added to a conventional microscopy slide. 1% of SeaKem LE agarose (Lonza) was added to 1 × M9 minimal medium (diluted from a 5× stock, Sigma-Aldrich) and heated until the agarose was completely dissolved. 95 µl of the solution was added into the GeneFrame chamber and the chamber sealed immediately with a conventional microscopy slide. Once set, the top slide was removed, and the agarose pad was air-dried for 20 min at room temperature and used immediately. Once the sample was added and air-dried, the GeneFrame chamber was sealed by adding a 22 × 22 mm cover slip. The visualisation was done using a Ti-U inverted microscope (Nikon) with a CFI Plan Fluor DLL 100 × objective (Nikon) and an ORCA Flash 4.0 LT plus camera (Hamamatsu). Phase contrast images were taken using a pE-100 single LED wavelength source

(CoolLED). For fluorescence, the pE-4000 illumination system (CoolLED) was used. The relevant filters for visualisation of CFP, YFP, NAO and mCherry were Zeiss filter sets 47 (CFP) and 46 (YFP/NAO), as well as Nikon TXRED-A-Basic (mCherry). Images were captured using the NIS Elements-BR software V4.51 (Nikon). For each experiment, three independent biological replicates were performed. Foci counts were generated manually, under conditions where processing was done blind in terms of test conditions. Postprocessing of images, such as cropping and rotating for presentation purposes, was performed in Adobe Photoshop CC (V24.0.0).

### Time-lapse microscopy

Fresh overnight cultures of the *E. coli* strains of interest were diluted 100-fold in fresh LB broth (Miller composition) and incubated with vigorous aeration at 37 °C until reaching $OD_{600}$ 0.2. 1 µl of the sample was pipetted onto an agarose pad and air-dried. For generation of pads 65 µl (15 × 16 mm) GeneFrames (Thermo Scientific™) were used. 1% of SeaKem LE agarose (Lonza) was added to LB broth (Miller composition) and heated until the agarose was completely dissolved. If required, saccharin was added to the molten agarose solution at the required concentration. For *E. coli* experiments, 95 µl of the solution was added into a GeneFrame and the chamber was sealed immediately with a conventional microscopy slide. Once the agarose had set, the top slide was removed and the pad air-dried for 20 min at room temperature and used immediately. For *A. baumannii* experiments, 5 GeneFrames were stacked on top of each other and added to a conventional microscopy slide. In all, 500 µl of the LB medium containing 1% of SeaKem agarose was added into the chamber of the stacked GeneFrames and the chamber sealed immediately with a conventional microscopy slide. Once set, the top slide was removed and 2 mm wide vents cut into the GeneFrame stack on all four sides. The agarose pad was then air-dried for 20 min at room temperature and used immediately. Once the sample was added and air-dried, the GeneFrame chamber was sealed by adding a 22 × 22 mm cover slip. Cells were visualised using the Ti-U system described above. For time-lapse imaging, the temperature was maintained at 37 °C using an environmental chamber (Digital Pixel). Time-lapse stacks were captured using the NIS Elements-BR software V4.51 (Nikon). Movie clips were directly exported to mp4 format. Postprocessing of images, such as cropping and rotating, was performed in Adobe Photoshop CC (V24.0.0).

### Bacterial permeability staining and fluorescence microscopy

Cell permeability staining was performed as previously reported in de Dios et al (2023). Briefly, *A. baumannii* AB5075 stationary phase overnight cultures were diluted 1/100 (v/v) in 15 mL of LB broth containing either 1% saccharin or an equivalent vehicle control in a 100 ml Erlenmeyer flask. Cultures were incubated at 37 °C, 180 rpm shaking for 2 h. After 2 h, 10 µl of a 1 mg/ml DAPI solution and 10 µl of a 5 mg/ml solution of Nile Red were added to each flask before incubating for 30 min at 37 °C, 180 rpm. Following incubation 10 ml of each culture was centrifuged at 5000 rpm for 5 min, and the supernatant discarded. Pellets were resuspended in 10 ml of sterile 4% formaldehyde in PBS and incubated statically in the dark for 30 min to fix the cells. Samples were then centrifuged at 5000 rpm and pellets were washed twice with 10 ml sterile PBS.

Washed pellets were then resuspended in 10 ml of sterile PBS and 10 µl of the cell suspension was spotted onto a glass slide and allowed to air-dry in the dark. Samples were imaged using Leica HF14 DM4000 microscope using CY3 (Ex: 542–568 nm, Em: 579–631 nm) and DAPI (Ex 325–375 nm, Em: 435–485 nm) filters. The native Leica Application Suite Advanced Fluorescence software (V4.0.0.11706) was used for image capture. Images were captured at ×200 magnification. Experiments were performed in biological triplicate and representative images are presented.

## Antibiotic uptake and flow cytometry

To measure the uptake of antibiotics after saccharin treatment, we used the fluorescently labelled derivative of penicillin, Bocillin (Bocillin™ FL penicillin, ThermoFisher Scientific), a fluorescently labelled derivative of vancomycin (BODIPY™ FL Vancomycin, ThermoFisher Scientific) and a customised fluorescently labelled derivative of neomycin (Neomycin-Cy5, synthesis procedure detailed)) in a flow cytometry-based approach. *A. baumannii* AB5075 overnight cultures were diluted 1/50 in 4 ml fresh LB medium and incubated for 1 h at 37 °C, 180 rpm. Cultures were treated with 1.5%, 1% and 0.5% saccharin, as well as a mock treatment during 30 min at 37 °C, 180 rpm. After this incubation, samples were harvested and centrifuged. After removing the supernatant, cell pellets were resuspended in PBS, and a final concentration of 5 µM of Bocillin, 5 µM of Vancomycin-BODIPY™ FL or 0.1 µM Neomycin-Cy5 was added (including an unlabelled control). Cell samples were mixed thoroughly and incubated at 37 °C in the dark for 10 min prior to flow cytometry detection. To estimate normal cell size, an unlabelled sample withdrawn from a stationary phase culture (non-dividing cells) was subjected to flow cytometry as well.

We performed flow cytometry using a BD FACSAria III Cell Sorter (2015-07-14 T, BD Biosciences) equipped with a red laser (633 nm), yellow-green laser (561 nm) and blue laser (488 nm). The blue laser was used to collect the forward scatter (FSC) and the side scatter (SCC) through a photodiode, and the Bocillin-FL/Vancomycin-FL signals through a 530/10 bandpass filter. The yellow-green laser was used to collect the Neomycin-Cy5 signal through a 710/50 bandpass filter. The threshold operator was set to 200 on SSC. The PMT voltages were customised for optimal separation of different populations. Instrument setting and data acquisition were performed using BD FACSDiva software (V 9.0). This software and FlowJo (V 10.8.1_CL) were used for data analysis. All samples were run at a flow rate of ~1000 events/s. Unlabelled overnight cultures (non-dividing cells) were used as cell size control, and unlabelled exponential phase cells were used to differentiate labelled and unlabelled populations (Appendix Fig. S11A,B).

Flow cytometry data was represented on a dot plot as Fluorescence-A (Y axis) against SSC-A (X axis), indicating probe uptake and cell size, respectively, using the FlowJo software. Normal size and oversized populations (X axis), as well as labelled and unlabelled populations (Y axis) were gated in the treated samples using the aforementioned controls as a reference.

## Antibiotic sensitivity assays

Antibiotic sensitivity tests were performed on Mueller-Hinton (MH) agar plates (pH 7.4) supplemented with $CaCl_2$ 2 mM and $MgSO_4$ 1 mM as previously described (de Dios et al, 2023). MH media was supplemented with 1.5% or 1% saccharin, or mock treatment, poured on 90-mm Petri dishes and left to air-dry. Once solidified, plates were inoculated with an *A. baumannii* AB5075 suspension in PBS (0.5 McFarland units) using a cotton swab. MICs for doripenem, imipenem and meropenem were measured using E-test strips (bioMérieux) according to the manufacturer's instructions. The minimum inhibitory concentration (MIC) values were read after incubating the plates at 37 °C for 24 h.

## Synthesis of Neomycin-Cy5

### *General procedures and materials for organic synthesis*

All reagents, obtained from Acros, Alfa, Sigma- Aldrich, TCI, Lumiprobe and VWR were used directly as supplied unless otherwise noted. Anhydrous solvents were dried by pre-storing over activated 4 Å molecular sieves. Thin-layer chromatography (TLC) was used to monitor reaction progress. TLC analysis was performed on precoated aluminium-based plates (TLC Silica gel 60 F254, Supelco), and plates were developed under UV irradiation (254 nm) or with $KMnO_4$ staining and subsequent heating. Preparative thin-layer chromatography was carried out using Merck TLC Silica gel 60 F254, size 20 × 20 cm. Column chromatography was performed with silica gel (Silica gel 60, irregular 40–63 µm for flash chromatography, VWR Chemicals). Automated flash column chromatography was performed with the automated Puriflash XS420 system from Interchim with an integrated UV and ELS detector and the specified solvent mixtures. Cartridges filled with C18-modified silica (Sfär C18 D–Duo 100 Å 30 µm) from Biotage® were used as the stationary phase. 1H NMR spectra were recorded at room temperature on a 400 MHz Bruker 9.4 Tesla Avance III HD system equipped with a SmartProbe (broadband). 13 C NMR spectra (1H decoupled) were recorded at room temperature on the same machine operating at 101 MHz. All chemical shifts (δ) are reported in parts per million (ppm) with internal reference to residual protons in CDCl3 (δ 7.26 or 77.16). Coupling constants (J) are given in Hz with an accuracy of 0.1 Hz. Splitting patterns are reported as broad singlet (bs), triplet (t), quartet (q) and doublet of triplet (dt). Low-resolution mass spectra (LRMS) were recorded on a Thermo Scientific LTQ Orbitrap XL instrument by direct injection of solutions of compounds in methanol, using electrospray ionisation in positive mode (ESI +). High-resolution mass spectra (HRMS) were recorded at the Faculty of Biosciences, Fisheries and Economics, UiT The Arctic University of Norway, using high-resolution Agilent 6540B quadrupole time-of-flight (Q-ToF) mass spectrometer with a dual electrospray ionisation (ESI) source, coupled to an Agilent 1290 Infinity UHPLC system, controlled by MassHunter software.

### *Synthesis and characterisation of the Neomycin-Cy5 probe*

The synthesis of Neomycin-Cy5 (SI-6) was performed according to the procedure of Sabeti Azad et al (2020).

**1,3,2′,6′,2″,6‴-Hexa-N-tert-butyloxycarbonylneomycin, SI-1:** To a solution of neomycin trisulfate (1.00 g, 1.10 mmol, 1.0 equiv.) in DMF/H$_2$O (1:1, 2 mL) was added dropwise at room temperature triethylamine (9.0 mL, 64.7 mmol, 58 equiv.), and then di-tert-butyl dicarbonate portionwise (2.10 g, 9.62 mmol, 9.0 equiv.). The mixture was heated to 60 °C for 5 h. After lyophilization, the crude

material was dissolved in ethyl acetate (40 mL) and washed with brine (4 × 15 mL). The organic layer was separated, dried over $Na_2SO_4$, filtered and the solvent was removed in vacuo. The residue was purified by column chromatography (10% methanol/$CH_2Cl_2$) to afford SI-1 (1.06 g, 0.869 mmol, 76%) as an off-white powder. LRMS (ESI +) calc. for $C_{53}H_{95}N_6O_{25}$ ([M + H] +) 1215.6, found 1215.3.

**5''-O-Triisopropylbenzenesulfonyl-1,3,2',6',2'',6'''-hexa-N-tert-butyloxycarbonylneomycin, SI-2:** To a solution of SI-1 (300 mg, 0.247 mmol, 1.0 equiv.) in anhydrous pyridine (3 mL) was added a solution of 2,4,6-triisopropylbenzenesulfonyl chloride (2.40 g, 7.93 mmol, 32 equiv.) in anhydrous pyridine (7 mL) at room temperature. The solution was stirred at room temperature for 68 h and was diluted with ethyl acetate (20 mL) and washed with water (3 × 15 mL) and with brine (15 mL). The organic layer was separated, dried over $Na_2SO4$, filtered, and the solvent was removed in vacuo. The residue was purified by column chromatography (10% methanol/$CH_2Cl_2$) to obtain SI-2 (175 mg, 0.118 mmol, 48%) as a white powder. LRMS (ESI +) calc. for $C_{68}H_{117}N_6O_{27}S$ ([M + H] +) 1481.8, found 1481.3.

**2-(N-Trifluoroacetamido)ethanethiol, SI-3:** The synthesis was performed in accordance with a literature procedure (Robins et al, 2010). To a suspension of finely powdered 2-aminoethanethiol hydrochloride (2.00 g, 17.6 mmol, 1.0 equiv.) in anhydrous toluene (27.1 mL) was added anhydrous pyridine (3.5 mL, 44 mmol, 2.5 equiv.). Trifluoroacetic anhydride (3.0 mL, 21 mmol, 1.2 equiv.) was slowly added, and the solution was stirred overnight at room temperature. The reaction was carefully quenched by the addition of $NaHCO_3$ (aq. sat., 20 mL. Warning: evolution of $CO_2$ gas). The phases were separated, and the aqueous phase was extracted with $CH_2Cl_2$ (3 × 20 mL). The combined organic layers were dried over $Na_2SO_4$ and filtered, and the solvent was removed in vacuo. The residue was redissolved in a minimal amount of diethyl ether/pentane (1:1) and filtered through a silica pad, eluting with 100 mL of the solvent mixture. The solvent was removed in vacuo, and the remaining material was dried under high vacuum to give amide SI-3 (2.62 g, 15.1 mmol, 86%) as a colourless oil.

1H NMR (400 MHz, CDCl3) δ 6.79 (1H, bs, NH), 3.56 (2H, q, J = 6.3 Hz, NCH$_2$), 2.75 (2H, dt, J = 8.6, 6.4 Hz, CH$_2$S), 1.41 (1H, t, J = 8.6 Hz, SH); 13 C NMR (101 MHz, CDCl$_3$) δ 157.5 (q, 2JCF = 37.1 Hz), 115.9 (q, 1JCF = 287.7 Hz), 42.6, 23.9.

Data are consistent with literature values (Robins et al, 2010).

**5''-(2-N-Trifluoroacetamidoethylthiol)-1,3,2',6',2''',6'''-hexa-N-tert-butyloxycarbonyl-5''-deoxyneomycin, SI-4:** To a solution of sodium ethoxide (2.8 M in ethanol, 0.91 mL, 0.012 mmol, 340 equiv.) was added thiol SI-3 in ethanol (1.4 mL) at room temperature and the mixture was stirred at room temperature for 15 min. Then a solution of sulfonate ester SI-2 (50 mg, 0.034 mmol, 1.0 equiv.) in ethanol (0.7 mL) was added at room temperature and the mixture was stirred at this temperature for 5 h. $CH_2Cl_2$ (5 mL) was added at room temperature and then an aqueous solution of potassium phosphate (pH 6, 15 mL) was added. The organic layer was separated, dried over $Na_2SO_4$, filtered and the solvent was removed in vacuo. The residue was purified by column chromatography (10% methanol/$CH_2Cl_2$) to afford SI-4 (22 mg, 0.016 mmol, 47%) as an off-white powder. LRMS (ESI +) calc. for $C_{57}H_{98}F_3N_7O_{25}NaS$ ([M+Na]+) 1392.6, found 1392.8.

**5''-(2-Aminoethylthio-)-1,3,2',6',2''',6'''-hexa-N-tert-butyloxycarbonyl-5''-deoxyneomycin, SI-5:** Amide SI-4 (22 mg, 0.016 mmol, 1.0

equiv.) was dissolved in a mixture of $NH_4OH$ solution (28% in water, 2.1 mL) and methanol (0.9 mL). The solution was stirred at room temperature for 4 h and the solvent was removed in vacuo to obtain SI-5 (20 mg, 0.016 mmol, quant.) as a white powder. LRMS (ESI +) calc. for $C_{55}H_{100}N_7O_{24}S$ ([M + H] +) 1274.7, found 1274.8.

**Neomycin-Cy5, SI-6:** The amine SI-5 (2.5 mg, 2.0 μmol, 1.0 equiv.) and Cy5 monofunctional NHS-ester (1.6 mg, 2.4 μmol, 1.2 equiv.) were dissolved in DMF (500 μmol) and N,N-diisopropylethylamine (10.0 μL, 57.4 μmol, 28 equiv.) was added at room temperature. The solution was stirred at room temperature for 30 min. The solution was diluted with chloroform for purification on a preparative TLC plate (2.5 mm, Silica gel 60; chloroform/ethanol/$NH_4OH$ (aq. 28%) 5.6/3.8/0.45). The silica gel was recovered and placed in a funnel with filter paper, and the product was eluted with ethanol (3 × 5 mL). For removal of the butyloxycarbonyl groups, the residue was dissolved in trifluoroacetic acid (90%, 1 mL) and the solution was stirred for 1 h at room temperature. The solvent was removed in vacuo and the residue was dissolved in aqueous HCl solution (10%, 2 mL). The solution was stirred for 15 min at room temperature, before volatiles were removed in vacuo. This procedure was repeated two times to afford the neomycin-Cy5 probe SI-6 (2.2 mg, 1.9 μmol, 95%) as a blue powder.

Tr 1.51 min; HRMS (ESI +) calc. for $C_{57}H_{88}N_9O_{13}S$ ([M] +) 1138.6217, found 1138.6447.

HPLC-HRMS data obtained for the generated neomycin-Cy5 (shown in Appendix Fig. S14) are consistent with literature values (Sabeti Azad et al, 2020).

## Preparation of saccharin-loaded hydrogels

To test the topical application of a saccharin formulation, we prepared hydrogels containing it as an active component. Potassium tetraborate (3%, w/w) and polyvinyl alcohol (3%, w/w), as well as sodium saccharin (6% or 8%, w/w) (no saccharin for the mock treatment hydrogels) were added into a beaker and mixed with deionized water. The mixture was placed in a water bath and heated at 80 °C for 3 h while being swirled with a stirring rod every 1 h. After obtaining the hydrogels, they were placed in a Petri dish and their shape was normalised with custom-made 3D-printed mould. The mould was 3D-printed in a Mars 2 Pro 2 K resin 3D printer (Elegoo). Freshly printed moulds were UV-cured in a Wash & Cure 2.0 chamber (Anycubic).

## Colony biofilm hydrogel treatment

In all, 1-ml LB agar plates were prepared in a 12-well plate. Overnight cultures of *A. baumannii* AB5075 were diluted to $OD_{600}$ 0.05 in PBS. In total, 5 μl of the diluted AB5075 cell suspension was applied to the surface of the agar and allowed to dry. The agar was incubated for 24 h at 37 °C to form a biofilm. After the incubation, saccharin-loaded moulded hydrogels with concentrations of 6% and 8% saccharin, a vehicle control and a silver-alginate wound care dressing were applied on the biofilm. An untreated biofilm was also used as a control. Then, the plate was transferred back to the incubator for a further 24 h. The hydrogel/dressing was removed, and the agar and the hydrogel surfaces were washed with 1 ml of PBS to resuspend the biofilm. The washing process was repeated three times with the same volume, diluting the collected bacterial suspension and enumerating viable cells.

**The paper explained**

**Problem**

Antibiotic resistance is making it more difficult to treat bacterial infections and as a result we are in urgent need of novel approaches to prevent or treat infections.

**Results**

We show that saccharin mediates its antimicrobial activity via bulge-mediated cell lysis rather than through impacting DNA replication dynamics. We reveal that saccharin can inhibit a range of virulence-associated phenotypes and that it can prevent and disrupt polymicrobial biofilms. We uncover the capacity of saccharin to potentiate antibiotics, increasing cell envelope permeability to antibiotics, ultimately overwhelming resistance mechanism. We demonstrate the therapeutic potential of a saccharin hydrogel in a porcine ex vivo wound model.

**Impact**

This study highlights the potential of saccharin to be developed as a novel antimicrobial treatment effective against multidrug-resistant pathogens.

## Ex vivo porcine skin assay

The porcine skin obtained from the pig belly was purchased from Fine Food Specialists Ltd, not frozen and free of additives. To mimic an acute burn wound, we followed the protocol described by Alves et al (2018) with modifications from de Dios et al (2023). Briefly, an array that contains 20 steel pins (8-mm diameter each) was heated to 140 °C for 1 h and then placed on a 10 cm² piece of porcine skin for 1 min. After inflicting the burns, the skin was cut into regular 1.5 cm² pieces which were placed in individual wells of a 24-well plate. Both sides of the skin sections were sterilised under UV light for 1 h. After sterilisation, the burn wounds were inoculated with 5 μl of an *A. baumannii* AB5075 suspension in PBS (OD$_{600}$ 0.05). After allowing the bacterial suspension to air-dry, the burnt pieces of skin were incubated for 3.5 h at 37 °C to allow the formation of a biofilm. Then, a moulded 6% saccharin hydrogel (prepared as described above) was applied on the burn wounds, and they were placed back into the incubator for 1 h. Following treatment, pieces of hydrogels were removed, and wounds were washed with 1 mL of sterile PBS while scraping the wound bed (3 times with the same volume of PBS) to recover the cells from the biofilm. The PBS was then diluted, and viable cells were enumerated.

## Statistical analyses

Graphs show either populational distributions of individual cell counts or average values ± SD (standard deviation). No blinding or randomisation was done to design the experiments. Data representation and statistical comparisons were performed on GraphPad Prism 10.1.2 (324) (GraphPad Software, San Diego, California USA, www.graphpad.com). The statistical analysis and *post hoc* corrections applied to each dataset, as well as the number of biological replicates performed per experiment, are indicated in the respective figure legends. Parametric or non-parametric tests were selected depending on the results of normality tests performed on each individual dataset. Exact *p* values for each statistical comparison are shown in Appendix Table S1 Sheldon and Skaar (2020).

## Data availability

RNA-seq data: Gene Expression Omnibus GSE276752 and GSE238183. 3D-printed hydrogel mould blueprint: NIH 3D Printing Repository 3DPX-020380.

The source data of this paper are collected in the following database record: biostudies:S-SCDT-10_1038-S44321-025-00219-1.

## Peer review information

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

## Acknowledgements

RRMC and EM are supported by the NC3Rs PhD Studentship NC/V001582/1. RRMC and RD are supported by a Biotechnology and Biological Sciences Research Council New Investigator Award BB/V007823/1 and an Medical Research Council Award MR/Y001354/1. RRMC and CP are supported by the Academy of Medical Sciences/the Wellcome Trust/the Government Department of Business, Energy and Industrial Strategy/the British Heart Foundation/Diabetes UK Springboard Award (SBF006\1040). RD received a Research Mobility Award from the Young European Research University Network to collaborate with CSL and visit UiT. The work was supported by joint Research Grants BB/T007168/1 and BB/T006625-1 from the Biotechnology and Biological Sciences Research Council to CJR and ELB, as well as BBSRC research grant BB/W000393/1 to CJR. The work was further supported by a grant by the North Norwegian Health Trust (HelseNord, HN 1688-23) to CSL. MMH and ALW would like to thank the Tromsø Research Foundation and UiT Centre for New Antibacterial Strategies (CANS) for a start-up grant (TFS project ID: 18_CANS).

## Author contributions

**Rubén de Dios**: Resources; Data curation; Formal analysis; Supervision; Funding acquisition; Methodology; Writing—original draft; Writing—review and editing. **Kavita Gadar**: Data curation; Formal analysis; Investigation; Methodology. **Chris R Proctor**: Data curation. **Evgenia Maslova**: Data curation; Formal analysis; Writing—review and editing. **Jie Han**: Data curation; Formal analysis. **Mohamed A N Soliman**: Data curation; Formal analysis; Writing—review and editing. **Dominika Krawiel**: Data curation; Formal analysis. **Emma L Dunbar**: Data curation; Formal analysis. **Bhupender Singh**: Data curation; Formal analysis. **Stelinda Peros**: Data curation; Formal analysis. **Tom Killelea**: Data curation; Formal analysis. **Anna-Luisa Warnke**: Data curation; Methodology. **Marius M Haugland**: Data curation; Formal analysis; Methodology; Writing—review and editing. **Edward L Bolt**: Data curation; Formal analysis; Supervision; Writing—review and editing. **Christian S Lentz**: Data curation; Formal analysis; Supervision; Methodology; Writing—review and editing. **Christian J Rudolph**: Conceptualisation; Resources; Data curation; Formal analysis; Supervision; Investigation; Methodology; Writing—original draft; Writing—review and editing. **Ronan R McCarthy**: Conceptualisation; Resources; Data curation; Formal analysis; Supervision; Funding acquisition; Validation; Investigation; Visualisation; Methodology; Writing—original draft; Project administration; Writing—review and editing.

Source data underlying figure panels in this paper may have individual authorship assigned. Where available, figure panel/source data authorship is listed in the following database record: biostudies:S-SCDT-10_1038-S44321-025-00219-1.

## Disclosure and competing interests statement

Brunel University London holds patents and PCTs covering the therapeutic use of artificial sweeteners and their use as antibiotic potentiators.

