## [Peer Review File · EMBO Molecular Medicine]

Saccharin disrupts bacterial cell envelope stability and interferes with DNA replication dynamics

Rubén de Dios, Kavita Gadar, Chris Proctor, Evgenia Maslova, Jie Han, Mohamed Soliman, Dominika Krawiel, Emma Dunbar, Bhupender Singh, Stelinda Peros, Tom Killelea, Anna-Luisa Warnke, Marius Haugland, Christian Lentz, Edward Bolt, Christian J. Rudolph, and Ronan McCarthy

Corresponding author: Ronan McCarthy (ronan.mccarthy@brunel.ac.uk)

Review Timeline:

Submission Date:	15th Jan 25
Editorial Decision:	18th Feb 25
Revision Received:	4th Mar 25
Accepted:	6th Mar 25

Editor: Zeljko Durdevic

Transaction Report: The first peer-review round of this manuscript was performed in August 2023 when the manuscript was rejected.

Response to Reviewers (Responses in Black)

Referee #1 (Remarks for Author):

The artificial sweetener saccharin is widely consumed and has previously been linked to alterations in the host gut microbiota. Given this it has been proposed that saccharin has the potential to be used as an antibacterial agent, however the mechanism of this antibacterial activity has not been investigated in detail. In this study the authors investigate how saccharin impacts the growth and biofilm-forming capabilities of specific bacteria with a focus on ESKAPE pathogens (mainly *A. baumannii* and *E. coli*). From the data presented the authors propose that saccharin impacts upon membrane integrity and interferes with DNA replication. Additionally, the authors demonstrate that saccharin can potentiate the action of several antibiotics *in vitro*. Finally, the authors test the impact of saccharin on *A. baumannii* growth in two *ex vivo* models. Overall I think the concept of using saccharin as an antibacterial and as an antibiofilm agent is interesting, however I do not think that all of the data presented in this manuscript supports the mechanisms of action for saccharin that the authors propose. I have described the major and minor specific issues I have in detail below.

Major points:

1. For the polymicrobial biofilms in Fig 3, how do the authors know that the biofilm they are treating is actually polymicrobial? Just because 3 different species were added at a 1:1:1 ratio into the LB doesn't mean that all species attach to the multi-well dish, indeed the authors mention that *S. aureus* doesn't grow as well in LB and so perform the same experiment again in TSB. Furthermore, without checking the composition of the biofilm the authors do not know if there are competitive interactions occurring to exclude one or more species, resulting in the actual biofilm only consisting of one species. It would be more convincing if the authors first presented data showing that they can recover each species from the untreated polymicrobial biofilm at a few time points eg a mid and end time point and determine relative amounts of each bacteria via CFUs on relevant selective media. Once they have determined that indeed they do have a polymicrobial biofilm then go on to test if saccharin could inhibit *de novo* biofilm formation and disrupt established biofilms.

The reviewer raises an excellent point here that we have sought to address with additional experiments and we have now included CFU counts for each species. Cells embedded in the biofilms were mechanically resuspended, serially diluted and plated on selective media for each species, including LB agar supplemented with gentamycin (20 mg/L), *Pseudomonas* isolation agar and mannitol salt agar for *A. baumannii*, *P. aeruginosa* and *S. aureus*, respectively (Supplementary Figure S6.), this has confirmed that all 3 species are represented in the 24-hour biofilm.

2. Fig 3A: why was 1% saccharin chosen as the concentration to try to inhibit *de novo* biofilm formation when 2% was the lowest concentration seen to inhibit *Acinetobacter* or *Pseudomonas* biofilms in Fig 2A? In Appendix Fig S1 a 2%

saccharin solution was used for the biofilm inhibition in TSB media and the decrease in biofilm is much more striking than 1% in LB in Fig 3A. Why was 2% used in TSB (Fig S1) and 1% in LB (Fig 3A)? The authors should use a consistent concentration of saccharin for the same kind of experiment unless there is a good reason to have different concentrations. The reasoning for these different concentration choices should be explained in the text.

We fully appreciate and understand the reviewers comment here and have now added greater rationale throughout the manuscript to clarify why each concentration was chosen. This concentration was selected to accurately determine what was a true anti-biofilm effect. Inhibiting the growth of these bacteria would also lead to a lack of biofilm formation, but due to the antimicrobial effect and not anti-biofilm effect. Therefore, to properly distinguish anti-biofilm from antimicrobial effects, we sought to test anti-biofilm properties of saccharin at the highest concentration that would not affect bacterial growth of the 3 species in the mixed culture as per the results shown in the current Fig. 5 which was 1%. To avoid redundancy, the experiments in TSB have been removed. It is also worth highlighting that concentrations as low as 0.1% did lead to a significant inhibition of *P. aeruginosa* biofilm formation (Figure 6B).

3. Lines 215-216: "The capacity to overcome this notoriously recalcitrant mode of multispecies growth highlights the therapeutic potential of saccharin in a clinically relevant situation." This statement is really overselling the results that were just presented. The data thus far is looking at the ability of bacteria to attach to a plastic multi well plate when grown in rich media - this is very different from showing any potential for saccharin to impact real world biofilms in a clinically relevant setting. The statement be toned down.

This has been removed.

4. Fig 4A: for the RNAseq analysis why was this done with 1% saccharin when there was no significant impact of this concentration on either growth (Fig 1) or biofilm formation (Fig 2)? It seems like it would make more sense to do the RNAseq at least 2% saccharin as this was the lowest concentration that was seen to impact growth of *A. baumannii* in Fig 1A? The authors need to explain the reasoning behind this concentration choice.

We apologize for any lack of clarity here, we have now explained the rationale as to why this concentration was chosen. Specifically, a sub-inhibitory concentration was chosen to give a greater insight into the anti-virulence effects of saccharin and the subsequent effect on gene expression without lethally impacting viability. Based on previous experience, performing transcriptomic analysis at concentrations close to the MIC typically leads to a higher non-specific background signal due to terminal phenotypes.

5. Lines 303-407: To investigate how saccharin may affect cell envelope stability the authors have chosen to perform time lapse microscopy with *E. coli* using 1.4% saccharin. I am struggling with the logic for this as to why *E. coli* has now become the bacterium chosen to investigate this when all data after Fig 1 has not included *E.*

coli and the idea that cell envelope stability may be impacted by saccharin (Fig 4) was based upon experimental data using *A. baumannii*? Additionally, the choice for using 1.4% saccharin to investigate the impact of this on *E. coli* is not explained - only 1% and 2% have been tested for growth inhibition (Fig 1A - ns at 1%, only significance seen at 2%, 1.4% saccharin data is not shown). The authors should show data on the impact of 1.4% saccharin on *E. coli* growth and biofilm development prior to using this % of saccharin in most of the microscopy that follows.

We apologise for the lack of clarity in the structure of the manuscript. We have now significantly restructured the text in alignment with this and the other reviewers' suggestions, such that the manuscript goes from the observations using *E. coli* as a model, due to the availability of established and validated genetic tools and reporters and its ease of use in the lab, to the general effect on a range of clinically relevant pathogens. To understand how the cell responds to saccharin exposure, we performed time-lapse microscopy using an *E. coli* model and treating with 1.4% saccharin, an effective concentration tested empirically in this setting and close to the theoretical half-maximal inhibitory concentration (IC₅₀) (Figure 1A, Supplementary Figure S1). Based on the reviewer's recommendation, we assessed all pathogens at 0.2% concentration increments within the 1-2% range, enabling us to determine respective IC₅₀ values. We decided to continue our study on the anti-biofilm and anti-virulence properties of saccharin with *A. baumannii* and *P. aeruginosa* and not *E. coli* because *E. coli* is a comparatively poor biofilm former and would not provide significant results on this matter (now shown in Sup. Fig. S5).

6. Microscopy in Fig 5. I have many issues with the data presented in this figure as outlined below.

a. There is no control for any of the time point data shown in Fig 5 - the authors need to show images of *E. coli* in the absence of saccharin grown for the same time periods with the same growth conditions, fluorescent dyes and eCFP-FtsZ strain to show that indeed the observed phenotypes are due to saccharin. How do the authors know that after 60 and 90 min of growth in the absence of saccharin that *E. coli* does not form membrane bulges in some cells, have multiple FtsZ rings and have chromosomal aberrations? In the absence of this control for each time point and sufficient fields of view imaged (see next point) the results presented cannot be interpreted.

We thank the reviewer for this insightful comment and we have now extensively rearranged this figure in accordance with these suggestions. Specifically, we have added the time-lapse of untreated *E. coli* cells, showing that bulges do not occur unless cells are treated. This phenotype has not been reported for *E. coli* in the absence of specific mutations or treatments that can trigger such membrane morphologies. We have also removed reference in the text and the figure to the FtsZ double ring structures as these were rare and less relevant to the overall conclusion that FtsZ rings appear to be forming.

b. For all microscopy performed the authors state that 2 biological replicates were performed on an unknown number of fields of view with a representative image presented. This implies that only 2 different fields of view were imaged - this is not a sufficient number of images to reliably make any conclusions about phenotypes. At least 10 fields of view should be taken for each condition and time point should be performed over 3 biological replicates. For this randomly selected fields should be chosen to image and not ones that have been cherry picked to have the phenotype the authors are looking for. Ideally the microscopy user should be blinded to the test condition they are imaging to remove the well known image capture bias.

We fully appreciate the reviewer's comment here and have worked hard to address this across the manuscript and I can now confirm we have performed at least three independent biological replicates and that all foci counting is done blind. Our original rationale stemmed from these assays being complementary and with each reporter and data set supporting the conclusions of the previous, therefore offering independent lines of evidence to support our conclusions. We hope the additional replicates, the foci quantifications, the dRNA-seq on *E. coli* and the additional flow cytometry experiments, which clearly show population-wide behaviours, address the reviewer's concerns.

c. In addition to the above points the authors should do image analysis on the multiple fields of view and biological replicates to ensure that any conclusions they make are not just a 'one off' - for example compare what cells look like +/- saccharin; what % of cells on average had bulges in the membrane +/- saccharin; what % of cells had more than one FtsZ ring +/- saccharin; what was the average length of cells +/- saccharin. This quantitative data should be presented graphically and accompany the actual images (either in the main figure or in the supplementary).

We appreciate the reviewers' comments here, have attempted to address them where feasible throughout the manuscript. Specifically, we have now included mock treated images across all figures with matching timepoints to the saccharin treated cells. We did not observe bulges in the membrane in any mock treated cells at any point and indeed this phenotype has not been reported for *E. coli* in the absence of specific mutations or treatments that can trigger such membrane morphologies. We have now removed reference to the double FtsZ ring formation from the revised manuscript as this was a rare phenomenon. Despite attempts, accurately determining cell length was unfeasible due to the impact of bulges on overall cell morphology. However, to address this point, we have determined the expected foci range based on maximum number of cell divisions that could have occurred during the imaging timeframe. The theoretical foci numbers that would follow a normal duplication progression have been highlighted in each figure in the Y-axis (dashed line) to give greater clarity to the reader as to how saccharin impacts foci numbers. We have also explained this in detail in the text.

d. There should be scale bars on the images in 5C - the legend says that they are not the same magnification as the image in 5B but just saying this with no scale on

the images in C is not acceptable.

These images have been removed for greater clarity.

7. Microscopy in Fig 6, 7, Fig S7 and S8: I have similar comments for these figures as for Fig 5 (point 6 above).

a. For each of these figures the "untreated control" is presumably images of *E. coli* prior to saccharin treatment. There should be a control for each time point (60 min for Fig 6A, S7 and S8, and 60 and 120 min for Fig 7) to be able to make appropriate comparisons and conclusions.

We apologise for any lack of clarity here, I can confirm that the mock control is present in all respective figures and that it is from a matching timepoint as the treated samples. This has now been labelled accordingly.

b. My second comment (6b) above is also relevant for these Figures - 2 biological replicates on an unknown number of fields of view is not sufficient, at least 10 fields of view (randomly selected, not cherry picked) should be imaged for each condition and time point and this should ideally be performed over 3 biological replicates. The analyses in Fig 6B, 6D should be then done on these additional fields of view and biological replicates. There should also be quantitative image analyses performed on Figures 7, S7 and S8 to be able to make appropriate conclusions on the biology and any comparisons between supplementary and main text figures.

Please see response to comment 6b.

c. For the legend in Fig 6 line 376-377 and in lines 359-360 the authors state that "Following treatment with saccharin, this pattern is dramatically altered, with cells showing significantly higher foci numbers" and "did not show the same significant increase in the number of foci when treated with saccharin", respectively. Where are the statistics to support the statements of significantly higher/not the same increase in foci numbers?

Thank you for this observation. We have now performed the respective statistical analyses for all foci quantifications and indicated them in each figure legend.

8. The data presented thus far on saccharin induced membrane damage in *E. coli* is not convincing to me as it is based upon just one image with one cell showing a small membrane bleb (Fig 5B and C) - this may become more convincing if what I have mentioned in point 6 above is addressed however, I would also suggest that other, relatively straight forward assays be performed to directly assess if saccharin impacts upon the membrane. For instance a fluorescence based assay that provides a quantitative output of membrane integrity like 1-N-phenyl naphthylamine (NPN) +/- saccharin on *E. coli*/*A. baumannii* at appropriate saccharin concentrations would provide more strength to the authors' claims that saccharin induces membrane damage.

We appreciate the reviewers' comment here and have performed several additional experiments to further validate our conclusions. To first confirm an increase in membrane permeability in *A. baumannii* by saccharin treatment, we performed fluorescence microscopy using Nile Red and DAPI, to which the cells are initially impermeable. After exposure to a sub-lethal 1% saccharin treatment, we could observe an increased DAPI uptake (Figure S10), thus further confirming the effect on membrane permeability. We have also performed additional flow cytometry analyses demonstrating the impact on the membrane. We believe these additional data in combination with our previous data (Figure 1: NAO staining of cells, Supplementary Tables S1-S4: Transcriptional evidence of a membrane response, Figure 8B: Cation supplementation increases cell viability, Supplementary Videos S1-S4) provide robust evidence of an effect on membrane integrity. We thank the reviewer again for these suggestions, however, as we are in no doubt that these additions considerably improve the manuscript.

9. The section beginning at line 409 onwards has now switched back to looking at *A. baumannii* and refers to the potential membrane damage seen in *E. coli* in the previous figures. Why have the authors switched back to *A. baumannii* now? I am again struggling with this logic as I mentioned above in point 5.

We have significantly reformatted the manuscript based on this and the other reviewers' comments. We fully appreciate the reviewers' feedback and believe we have significantly improved the narrative of the manuscript as a result.

10. Figure S9. The authors want to see if saccharin impacts on membrane permeability in *A. baumannii*. I do not understand why the authors have used DAPI as the stain to determine if this is occurring - DAPI is membrane permeable and stains AT rich regions of DNA, and will cross the membrane of essentially all bacterial cells, live or dead. I don't believe that there was no DAPI signal in the untreated *E. coli* sample, this just doesn't make any biological sense. I would suggest that the authors should instead show saccharin impact on membrane permeability in a more direct and standard manner that, for eg. using NPN (as above in point 8). I am also unsure why the authors used Nile Red here - it actually looks like these images highlight that the focal plane in the control image panel is not in focus with the cells, as it is in focus for the 1% saccharin cells. Is that why no DAPI signal was detected as the focal plane was not inline with where the cells actually were? Also it is also not clear to me why 1% saccharin was used here as this % does not have any significant impact on *A. baumannii* growth (Fig 1D) and the microscopy in previous figures with *E. coli* used 1.4% saccharin.

Please see response to comment 8 above. We have now introduced additional experiments that demonstrate the impact of saccharin on the cell envelope permeability, including a range of saccharin concentrations in a flow cytometry setup that shows a dose-dependent response in terms on cell morphology and permeability to different fluorescently labelled antibiotic probes (Supplementary Figure S11). The response to the next comment provides further details on these experiments.

11. Lines 424-426: " The proportion of cells with an increased size ranged from a 15.3% in the untreated control to a 53.1% in the case of cells treated with 1.5% saccharin (Appendix Figure S10C), which is coherent with our previous microscopy results". There is no quantification of cell size from microscopy data to allow this statement to be convincing. This quantification should be done as mentioned in my points 6c and 7b above, then these cell size comparisons can be done between the flow cytometry and microscopy data to support this statement.

Thank you for this comment, I can confirm we have performed several additional experiments to fully address this concern. In flow cytometry, cell size is measured independently of fluorescent labelling (we have now explained this in more detail in the respective Methods section). We did attempt to quantify cell length also in the microscopy however this was not possible due to the bulge formation. However, we have assessed cell morphology changes in a saccharin dose-dependent manner with two additional fluorescently labelled antibiotics to further address this comment. Flow cytometry analysis with vancomycin fluorescent derivatives confirmed our findings with Bocillin, with saccharin causing altered cell size and increased permeability to the antibiotic probes. As Bocillin and Vancomycin have cell envelope associated targets, we wanted to further challenge our hypothesis by assessing the impact of saccharin on the permeability to an antibiotic that targets an intracellular process such a ribosomal function. To this end we synthesized a bespoke neomycin probe (neomycin inhibits the ribosomal function) by conjugating it with Cy5. Flow cytometry analysis demonstrated a similar trend as to what was observed with Bocillin and Vancomycin, confirming that each antibiotic could access the cytosol in higher proportions in the presence of saccharin, overcoming this first protection barrier (Figure 9 and Sup. Fig. S11). We thank the reviewer again for these suggestions as we are in no doubt that this additional data considerably improves the manuscript.

12. From lines 461-476 and Fig 9A-B. The authors propose a novel infection model, the *ex vivo* caesarean section model which uses mouse cadavers. I think that the claims about this model might be somewhat oversold. Can this really be called an infection model since there will not be any immune system response to the applied bacteria from the (dead) host?

We have removed this model from the study based on the reviewer's suggestion that this is not an infection model.

13. From lines 479-496 and Fig 9C-D. The authors test the impact of a saccharin loaded hydrogel on a colony biofilm and on an *ex vivo* burn wound model on porcine skin. The colony biofilm used here was generated by spotting 5 ul of an overnight culture onto an agar plate and incubating at 37 C for 3.5 hrs (lines 892-897). Similarly the biofilm on the burned porcine skin was allowed to form for 3.5 hrs (lines 911-914). 3.5 hrs is a very short amount of time and I would be surprised if a colony biofilm or a biofilm on burned porcine skin had actually formed in this amount of time. There are clearly some cells attached to the surface of the agar or porcine skin to be

recovered for the CFU outputs but have the authors performed any validation that there is indeed a biofilm formed in both scenarios? The authors make a very strong claim about the potential of using saccharin to treat biofilm-associated wound infections (lines 494-496): "Together with the previous results, this further reinforces the therapeutic potential of saccharin and its capacity to tackle a major clinical burden as biofilm-associated wound infections". Until the authors can demonstrate that there is indeed a biofilm formed in their model I do not think that the data presented here supports the claim that saccharin has any capacity to target biofilm-associated wound infections and should be revised.

We appreciate the point the reviewer is raising here and it does come back to a fundamental definition of when a biofilm is a biofilm. Cells attaching to a surface is widely considered stage one of the biofilm life cycle and as such, our assays (both agar and porcine skin) confirmed that cells were at least in this first stage of biofilm formation. As correctly pointed out by the reviewer, we could recover colonies, confirming that cells had indeed attached to both surfaces. We do appreciate however that this most likely represents the early stages of the biofilm life cycle. To make this clearer to the prospective reader we have edited the text to refer to early-stage biofilms. Furthermore, to fully address this comment we repeated the colony biofilm assays after 24 hours of biofilm maturation. As with the assays at 3.5 hours (moved to Sup. Fig. S13), treatments with saccharin-loaded hydrogels lead to a significant reduction in viable bacteria in a 24-hour old biofilm (Figure 10A). We again thank the reviewer for this suggestion as we are in no doubt it has improved the strength of our overall conclusions.

Minor points:

1. Line 179 - "a striking 81%..." is reported in legend for Fig 2C to be 81.21% - the number of decimal places should be consistent when reporting on the same data. Other %s are reported to 1 decimal place throughout so perhaps best to stick to this for all %s reported in general.

This has been corrected throughout.

2. Lines 738-740: "For the polymicrobial MBIC experiments cultures of *A. baumannii* AB5075, *P. aeruginosa* PA14 and *S. aureus* CCUG 68792 were co-inoculated in a 1:1:1 ratio into LB containing either 2% or 4% saccharin in a 96-well plate" - Fig 3A reports 1% saccharin used for the LB biofilm inhibition of the mixed biofilms, I cannot see 2 and 4% reported for LB mixed biofilms anywhere. Also were the TSB mixed biofilm inhibition experiments (Fig S1) performed in the same way as the LB mixed biofilm inhibition ones? I cannot see the information for TBS mixed biofilms inhibition (only disruption) in the methods section. Similarly I cannot see methods for how the mixed LB biofilm disruption were performed. The only methods I can see for mixed biofilm disruption is in lines 743-746 where TSB is used initially to grow the mixed biofilm, then this is removed and LB + saccharin added - I assume this is meant to be TBS + saccharin that was applied otherwise this doesn't make sense that the kind of media was switched half way through the experiment.

The methods section has been updated with more detail. The experiments in TSB have been removed from the current manuscript version to avoid redundancies.

3. Line 778: Perhaps something like "single-cell fluorescent microscopy" is more appropriate than "single-image microscopy".

This has been adjusted.

4. How were image analyses performed for the data presented in Fig 6B and D? This should be described in the methods section.

This has been corrected.

Referee #2 (Remarks for Author):

The manuscript by de Dios et al investigated the antimicrobial mechanisms of saccharin and used a variety of different methodologies to determine the effect on virulence in vitro and ex vivo. There are a lot of experiments and most of them make sense, but the manuscript reads like as there are two separate stories being told and it would benefit from separation (first part about *A. baumannii* and second using *E. coli*). Also, given that saccharin showed better activity against *P. aeruginosa*, it is not clear why the focus was mainly on *A. baumannii*. I also read their earlier publication on the artificial sweetener ace-K, which sounded like a very similar story, and therefore limited my excitement about the novelty of the described findings.

My overall concern is that the hyperactivation of DNA replication (in *E. coli*) does not seem to overlap with the data from the RNA-Seq experiments (in *A. baumannii*). A similar disconnect appears to be present for defects in the nucleoid conformation or DNA damage as transcriptome sequencing does not hint towards that. Moreover, I not convinced that there is any drug interaction based on the provided experimental data and overall the entire work lacks controls. The manuscript would be stronger if the mechanism of bulge-mediated cell lysis could be proven for all investigated organisms including Gram-positive strains. Some evidence/mechanism of how saccharine ends up in the cell wall would be beneficial as well. Many parts of the introduction feel distracting and misplaced and should be deleted; other parts almost read like a copy of the Wikipedia page.

We sincerely appreciate the reviewer comments here and have completely reformatted the manuscript as a result. We would like to thank the reviewer for their suggestions as we now feel as such we have written a much more cohesive narrative. We also have made a greater effort throughout the manuscript to highlight how this study is novel with respect to the previous study which explored a structurally unrelated sweetener ace-K including highlighting how ace-K does not impact DNA replication dynamics (Figure 1-5, Supplementary Figure 2-4) and the successful integration of saccharin into a bioactive hydrogel that displays greater antibacterial activity than a market leading silver alginate wound dressings (Figure

10, Supplementary Figure 13).

Major comments:

- The introduction is not fit for purpose and needs to be rewritten to be tailored to the story told in this manuscript. Ln64-79: This entire paragraph is unnecessary to introduce as it does not fit the purpose and aim of this manuscript. Ln80-104: The impact on the microbiome is certainly interesting but has nothing to do with the work presented here and should therefore be deleted.

The introduction has now been reformatted significantly and in accordance with guidance from this reviewer. We have kept some references to the impact of saccharin on the microbiome as we feel it is important to contextualize our work with respect what is already known about the sweetener from a microbial context. However, we have abridged this section significantly compared to the original.

- Figure 1: There is no indication of what specific strains were used or what the experimental conditions were. It is overall confusing as different strains are mentioned in the manuscript - e.g., for *E. coli*, it is NCTC13476, AB1157, MG1655 etc. The presented data points are endpoint measurements and a better representation would be actual colony counts rather than optical density. In fact, the experimental setup basically shows an MIC experiment and it would be better represented as such (in addition to bacterial survivors). Are bacterial cells still alive when treated with the high saccharin concentrations? The y-axis should read Absorbance (OD600). Why does PA14 reach barely an OD of 1.1 after 18 h in LB? While I understand that some data has been published previously, similar growth experiments with the new strains and artificial sweeteners and other controls would be beneficial. The mechanism of saccharin appears very similar to that of sucrose - so a sucrose control (or other sugar) would be worthwhile to include. Can bacteria metabolize saccharin?

This figure has now been moved to much later in the manuscript (now Figure 5) and we have now provided additional clarity on the strains used in the figure legend. Furthermore, we have now included IC50 values (Sup. Fig. S1) to provide additional information and resolution on the inhibitory effect of saccharin on growth. PA14, as well as the other strains used, does not reach much higher optical densities because of the experimental setup in 96-well plates, which limits the humidity and aeration as compared to a conical flask or test tube. We have modified the figures to show absorbance as advised.

While sucrose has been shown to inhibit bacterial growth as highlighted by the reviewer, these are at concentrations significantly higher than what have been used in this study (>50g/100ml) (PMID: 6870223). We have not previously published MIC data relating to saccharin or the range of bacteria whose growth it can inhibit. The reviewer raises an interesting point about the metabolism of saccharin but currently there are no published reports of saccharin being metabolised by bacteria or the human body (PMID: 31601053).

- Biofilm experiments: Ln171: biofilm inhibition is interesting for Pa not so much for

Ab as it is probably just a matter of growth inhibition (2% kills most cells, hence no biofilm). Figure 2C seems to confirm that saccharin has no real anti-biofilm activity against Ab - although a statistically significant reduction, this is much more above the inhibitory concentration and therefore probably not really reflective of real anti-biofilm activity against this strain. What is the variation of the control biofilm mass in Figure 2C?

Thank you for this comment, we have softened our conclusions on the de novo biofilm inhibition potential of saccharin on *A. baumannii*. However, we disagree on the effect of saccharin on mature biofilms (current Figure 6C). Although it is true that the saccharin concentration is above those measured as MIC and MBIC, a mature biofilm is a lot more recalcitrant than single cells starting the biofilm formation process. Thus, we believe that this observation is valuable to assess the potential of saccharin to remove already formed biofilms. This has been clarified in the text. We have also added additional data where saccharin augmented hydrogels were assessed against 24-hour old biofilms, demonstrating a significant reduction (Figure 10A). The control biofilm mass variation was within the same range as that for Figure 6AB 0% controls.

- Ln197-217: While these experiments are certainly relevant, there is a lack of evidence that these are indeed polymicrobial biofilm conditions. Without proper CFU counts, these biofilms can be driven by any of the involved species and the effect of saccharin rather misleading as there might be one or the other dominant species within the biofilm mixture. *P. aeruginosa* is well known to overtake *S. aureus* in mixed biofilms.

The reviewer raises an excellent point here and we have now included CFU counts for each species. Cells embedded in the biofilms were mechanically resuspended, serially diluted and plated on selective media for each species, including LB agar supplemented with gentamycin (20 mg/L), Pseudomonas isolation agar and mannitol salt agar for *A. baumannii*, *P. aeruginosa* and *S. aureus*, respectively. While as the reviewer has highlighted that *P. aeruginosa* does begin to overtake *S. aureus*, at the time point selected (24 hours) in this experimental setup, *S. aureus* still represents a significant proportion of the biofilm.

- Ln235/253: What was the rationale of using a FC of 1? Given that low FC, qRT-PCR experiments would be needed for further verification. Using a higher FC would certainly overcome this, but then only a few genes would be dysregulated and their meaning rather biologically minor. Gene enrichment is probably not relevant due to the small number of genes dysregulated. Is Appendix Fig S2 the same as Fig 4A?

Yes, the original Fig S2 was the same as Fig 4A with one outlier removed to aid with visual representation. We have highlighted this in the figure legend for Figure 4A (now Figure 8A in current version of manuscript). We apologise for any lack of clarity, this is a Log_2FC of 1 (which amounts for an FC of 2) rather a FC of 1.

- Ln239-250: The treatment with sub-inhibitory concentration of saccharin (1%) did not affect biofilm inhibition (Fig 2A). How is the description of these genes relevant if it does not match the observed phenotype. The mention about 'coherent' (Ln248) is wrong and the given reference confusing. What was the FC of these individual genes? There is a lack of details.

We have now removed the reference to biofilm in this section and highlighted the link to motility more clearly.

- Ln303: The switch to *E. coli* out of a sudden is confusing and totally disrupting the flow of the manuscript. There is only one example shown in Fig 5A - how do all the other cells look like? A proper control is missing. It is also confusing whether, or if at all, the *E. coli* phenotype translates to *A. baumannii*, as the RNA-Seq data showed only minor changes (although similar MIC). Based on the evidence of filamentation and membrane bulges (in *E. coli*), one would expect that cells are highly stressed and therefore express a complete different transcriptional landscape - but this appears not to be the case. The *ftsZ* studies further disconnect the entire manuscript that was so far focused on *Ab* - and again, chromosomal aberrations should lead to massive transcriptional dysregulation. The 'ghost' phenotype is not very convincing unless further analyzed with confocal imaging (e.g., evidence using 3D stacking).

We appreciate these comments by the reviewer and have now restructured the manuscript significantly based on this to create a more coherent and progressive narrative. We have now added additional controls to all of our microscopy data and performed additional replicates which all add further strength to our conclusions. We have also added foci quantification and statistics to help support our conclusions. To sharpen the clarity and focus of the manuscript we have removed reference to the *FtsZ* double rings and have softened our interpretation on the impact to septum formation. The "Ghost" cell phenotype is simply a term to refer to the remaining cellular body post lysis, we have added additional negative controls to better reflect this distinction. We have now undertaken an additional RNA-Seq experiment comparing the *E. coli* strain used in the microscopy experiments in the presence or absence of 1.4% saccharin, in which pathways associated with the cell envelope and DNA replication and mismatch repair were all significantly differentially regulated.

- Ln430-440: The statement of antibiotic potentiator is not well justified. At a sub-inhibitory saccharin concentration of 1% there was no combinatorial effect. However, at a concentration of 1.5%, which probably killed most of the bacteria, a drastic effect was found. This indicates that the effect was likely driven by saccharin only and not a combination. Synergy experiments (e.g., checkerboard) would be needed to confirm, but most likely there is no interaction at all. Flow cytometry: How many dead cells were in this mixture?

The reviewer raises an interesting point here and we apologize for any lack of clarity in our wording. We agree that the critical inhibitory saccharin concentration is around 1.5%, which is coherent with the IC50 values we present in Sup. Fig. S1 for *A.*

baumannii among other species, with 0.2% concentration increments within the 1-2% range. To note, the experiments presented in Fig. 5, as well as those in Fig. 9A, B, were performed used in LB broth. However, to follow the EUCAST antimicrobial sensitivity testing guidelines, we did the carbapenem MIC tests in Fig. 9C and Sup. Fig. S12 using cation-adjusted Mueller-Hinton agar. This medium supported a better growth of *A. baumannii* in the presence of saccharin compared to LB, which allowed growth of a bacterial lawn in the presence of 1% and 1.5% of the sweetener, as it can be observed in the pictures displayed in Sup. Fig. S12, which shows a lawn-type growth in the areas that are not reached by the inhibitory concentration of carbapenems. For this reason, in this particular assay, we considered 1% and 1.5% saccharin as sub-inhibitory concentrations, and the growth observed in the presence of carbapenems as an antibiotic potentiation due to the membrane permeabilization showed in Fig. 9B. Now we have clarified this in the text (L560-562).

To further address this comment however we firstly assessed another antibiotic (Vancomycin, a glycopeptide that targets peptidoglycan precursors). Flow cytometry analysis confirmed our findings with Bocillin, with saccharin causing altered cell size and increased permeability to antibiotics. As Bocillin and Vancomycin have cell envelope associated targets, we wanted to further challenge our hypothesis by assessing the impact of saccharin on the uptake of an antibiotic that targets an intracellular process such a ribosomal function. To this end we synthesized a bespoke neomycin probe (neomycin inhibits the ribosomal function) by conjugating it with Cy5. Flow cytometry analysis demonstrated a similar trend as to what was observed with Bocillin and Vancomycin, confirming that each antibiotic could access the cytosol in higher proportions in the presence of saccharin, overcoming this first protection barrier (Figure 9). We thank the reviewer again for these suggestions as we are in no doubt that this additional data considerably improves the manuscript.

Other comments:

- Ln142: it is not really in a dose-dependent manner (except for Pa) - often just a yes or no growth answer.

We completely agree with the reviewer on this point as it was something we recognized ourselves. To gain further insights on this and a better resolution of the antimicrobial potential of saccharin, we measured its antimicrobial activity using 0.2% concentration increments ranging from 0-2% for *A. baumannii*, *E. coli*, *K. pneumoniae* and *S. aureus* (Lines 358-362, Sup. Fig. S1). This showed an IC50 ranging from 1.2-1.5%. In the case of *P. aeruginosa*, for which we could use a similar concentration range as in Figure 5 to calculate the IC50, we calculated this parameter as 2.5%, which indicated that this pathogen has a better tolerance to the treatment compared to the others tested.

- 147-151: This is all speculation with no evidence and should be deleted.

This has now been deleted.

- Ln168-169: Why are these references listed here? These do not seem appropriate to be listed here.

These citations were included to support the claim that both pathogens are strong biofilm formers. (Mulcahy et al., 2014, Maslova et al., 2021, Harding et al., 2018).

- Ln179: 'striking' might not be the correct wording unless proven with some controls. How would an antibiotic perform in comparison?

We have now removed the word striking.

- Ln251: What about other pili-dependent motility phenotypes (e.g., swarming)?

Thank you for this comment. We tested twitching as it was directly related to the downregulation of *pil* genes in the presence of saccharin showed in the respective RNA-seq dataset and served as a means of validating our transcriptomics. However, the only other motility mechanism documented in *Acinetobacter baumannii*, which is surface-associated motility (formerly known as swarming), has been demonstrated to be pili-independent (PMID: 22752907). For this reason, we decided not to test it.

- Ln265-277: Use either the word 'saccharine' or 'saccharin' throughout the manuscript. The fact that 1.3% saccharin caused a growth defect (S5A) is of concern as this is not obvious from the main figure (Fig 1D) - is there a concentration dependency between 1 and 2%? How were the concentrations for these metabolites chosen? Has a combination of all of them been considered?

We have removed this section for clarity. We have also checked to ensure that the same spelling is used for saccharin throughout.

- Ln287-289: This is a far stretch to say given the very small change! Was the difference only seen when calcium and magnesium were combined.

We have softened our conclusions on the effect of cations on saccharin inhibition. However, we believe that our interpretation points in the right direction, as it is recognised that cation supplementation can only partially relieve envelope-damaging effects of antimicrobials, even if modestly as for saccharin, by bridging and stabilizing the LPS layer (LOS in the case of *A. baumannii*), however they cannot completely prevent cell lysis (PMID: 25489959, PMID: 23103254). This is further supported by the fact that a monovalent cation such as potassium does not produce the same effect (Sup. Fig. S8).

- Ln377: What is strain JD1708?

A full table of strains has now been included (Supplementary Table S6). This is a derivative of *E. coli* MG1655 bearing the pTK135 plasmid (encoding the Cas1-LFP-Cas2 reporter). This is clarified in the Materials and Methods section.

- Ln416-418: This needs to be further quantified to draw any conclusions.

Thank you for this suggestion. To further address this comment we firstly assessed another antibiotic (Vancomycin, a glycopeptide that targets peptidoglycan precursors). Flow cytometry analysis confirmed our findings with Bocillin, with saccharin causing altered cell size and increased permeability to antibiotics. As Bocillin and Vancomycin have cell envelope associated targets, we wanted to further challenge our hypothesis by assessing the impact of saccharin on the uptake of an antibiotic that targets an intracellular process such a ribosomal function. To this end we synthesized a bespoke neomycin probe (neomycin inhibits the ribosomal function) by conjugating it with Cy5. Flow cytometry analysis demonstrated a similar trend as to what was observed with Bocillin and Vancomycin, confirming that each antibiotic could access the cytosol in higher proportions in the presence of saccharin, overcoming this first protection barrier (Figure 9, Sup. Fig. S11). We thank the reviewer again for these suggestions as we are in no doubt that this additional data considerably improves the manuscript.

- Ln468: What other wounds are referred to here and why do the authors think this wound is so much different to any other wound that involves surgery (sutures)?

This section has been removed in the current version of the manuscript after reviewers' feedback.

- Ln474: What was the rationale for 1 h treatment? How many wounds were investigated? Did every silk have the same thread length (as this might affect colonization)? How many bacteria actually attached to the suture rather than the wound?

We appreciate the reviewers' comments here on this assay and given the strength of the other ex vivo assays we have removed this assay from the study.

- Ln487: What is the evidence that Ab is in a colony biofilm? There is no reference for these experiments and I wonder how relevant this is in terms of biofilm production given the short timepoint.

We appreciate the point the reviewer is raising here and it does come back to a fundamental definition of when a biofilm is a biofilm. Cells attaching to a surface is widely considered the first stage of the biofilm life cycle and as such, our assays (both agar and porcine skin) confirmed that cells were at least in this first stage of biofilm formation, as we could recover colonies at 3.5 hrs, confirming that cells had indeed attached to both surfaces. We do appreciate however that this most likely represents the early stages of the biofilm life cycle. To make this clearer to the

prospective reader we have edited the text to refer to early-stage biofilms. However, to fully address this comment we repeated the colony biofilm assays after 24 hours of biofilm maturation. As with the assays at 3.5 hours (moved to Sup. Fig. S13 in the current version), treatments with saccharin-loaded hydrogels lead to a significant reduction in viable bacteria in a 24-hour old biofilm (Figure 10A of the current version). We again thank the reviewer for this suggestion as we are in no doubt it has improved the strength of our overall conclusions.

- Ln492: How many samples were used?

Data shown represents the average of three biological replicates \pm S.D.

The following comments largely focused on the discussion which has been significantly re-written in light of the experiments undertaken during the revision of the manuscript.

- Ln 517: How much 'free' saccharin would reach the gut? What percentage and how would that translate to the concentrations used here?

The concentrations used here far exceed the concentrations that could realistically be achieved in the gut with the RDI for saccharin being 5mg/kg. This is why we focused on topical therapeutic potential.

- Ln524-527: Did the authors use all those controls against all the pathogens mentioned in this study, in their previous study? If not, experiments with the remaining pathogens should be performed to make such statements.

Yes the same strains were used, in fact we tested several additional strains of *S. aureus* since publication of the original Ace-K study and have always noted limited inhibitory activity.

- Ln535: What is the concentration of saccharin in wastewater?

This varies but is typically below 0.80 $\mu\text{g/L}$ (Pang et al., 2020).

- Ln582-583: This is an interesting statement, as looking at the RNA-Seq data in their previous publication, would indicate the opposite - ie, ace-K could have a more pronounced effect on DNA replication due to the high number of dysregulation of genes.

We do agree with this reviewer that the specificity of this phenotype to saccharin and not ace-K is intriguing. Coherently with this observation, we could not identify any gene directly associated to DNA replication in our previous dataset using ace-K.

- Ln590: The authors could easily perform whole genome sequencing to obtain further evidence.

The reviewer raises an interesting point, but we wanted to emphasise the antimicrobial effect of saccharin and its clinical potential in this piece of work.

- Ln655: What happens when saccharine breaks down? What is the half-life and can bacteria metabolize it? In the end, does it offer a source of nutrients in the form of carbon sources?

Saccharin cannot be metabolized by the body (PMID: 31601053) and currently there are no reports of bacteria being capable of breaking it down.

- Ln707: based on Figure 1, this should start with 0.1%

This has now been adjusted.

- Ln762-766: add version numbers for all packages and scripts. Are the individual scripts available online? It is not clear whether the p-value or the adjusted p-value has been used? It would be good to provide PCA plots in the supplements.

The methods have been updated based on these comments and the tables updated with the p-values and the adjusted p-values.

- Ln870: What does 'regularly applied' mean?

This assay has now been removed.

- K. pneumoniae is spelled wrong throughout the manuscript.

This oversight has now been amended.

Referee #3 (Remarks for Author):

This manuscript has revealed new roles of saccharin in bacterial membrane stability and DNA replication, and this implies that saccharin could be used treat MDR bacterial infection. However, there are some concerns needed to be addressed.

1. This manuscript has demonstrated that saccharin could interfere the DNA replication in bacteria. But the results are very limited. There should be more molecular evidences to prove this finding like purifying the interactive proteins with saccharin and TEM structure. Moreover, the RNA-seq data seems not find the enriched genes for DNA replications. Also, for safety reason, the role of saccharin in DNA replication could be found in mammalian cells?

We have now added several additional data sets to support the conclusion that there is a genuine effect on DNA synthesis, however we now also acknowledge that the impact of these altered DNA replication dynamics is likely non-lethal as it is

superseded by the potency of the impacts of saccharin on the cell envelope, we have therefore softened our conclusions to reflect this.

To gain further insights on the effect of saccharin on DNA synthesis using the same model as we used to obtain the microscopy evidence of this (*E. coli*), we performed a dRNA-seq experiment following the same bacterial growth and saccharin treatment protocol we used for the prior microscopy experiments (GSE276752; reviewer access token: kxwncqysnpqqlbmj). KEGG pathway analysis of this dataset identified various functional gene clusters, including DNA replication (eco03030) and mismatch repair (eco03430), as significantly upregulated upon saccharin exposure. This further supports an effect of saccharin on DNA replication/synthesis. We have also quantified the impact of ace-K, another sweetener that induces cell filamentation and membrane damage, but DNA synthesis alterations are much more limited than those observed for saccharin, even at higher concentrations than those used for saccharin, therefore the effect on DNA synthesis dynamics appears specific to saccharin (Supplementary Figure S4). We also show that increasing the concentration of saccharin has a proportional impact on DNA synthesis (Figure 3). It is interesting to note that the effects on DNA synthesis did not appear in the RNA-Seq associated with *A. baumannii*. This could be due to species-to-species differences in the saccharin sensitivity, whereby DNA replication effects are masked in *A. baumannii* or occur earlier upon exposure. We now highlight this in the discussion.

2. Whether Saccharin at the dosages of inhibiting the proliferation of bacterial growth can destabilize the cell membrane and increase the permeability.

We appreciate the reviewers' comment here and have performed several additional experiments to further validate our conclusions. To first confirm an increase in membrane permeability in *A. baumannii* by saccharin treatment, we performed fluorescence microscopy using Nile Red and DAPI, to which the cells are initially impermeable. After exposure to a sub-lethal 1% saccharin treatment, we could observe an increased DAPI uptake (Figure S10), thus further confirming the effect on membrane permeability. We have also performed additional flow cell analysis demonstrating the impact on the membrane. We believe these additional data in combination with our previous data (Figure 1: NAO staining of cells, Supplementary Tables 1-4: Transcriptional evidence of a membrane response, Figure 8B: Cation supplementation increases cell viability, Supplementary Videos 1-4) provide robust evidence of an effect on membrane integrity. We thank the reviewer again for these suggestions, however, as we are in no doubt that these additional data considerably improve the manuscript.

3. Is it possible that MDR bacteria develop resistance to saccharin? Have the authors tested this?

This is an excellent suggestion and one that we explored extensively using a range of different assays including serially passaging *A. baumannii* at 2-fold increasing saccharin concentration assays beginning with a sub-MIC over time (PMID: 38247653), growth on agar slant plates with increasing concentrations of saccharin

from one side of the plate to the next (PMID: 37117027). However, none of these assays resulted in the production of strains with increased resistance. We also screened the entire AB5075 transposon mutant library (PMID: 25845845) for mutant strains that could grow at >3% saccharin and could not identify any resistant mutant. We are happy to include this negative data in manuscript should the reviewer think it would be useful, however for now we have excluded it as it is entirely negative results.

4. In the animal model, there is a lacking a positive control to compare the effectiveness of saccharin hydrogels.

Due to recommendations from the other reviewers we have removed the C-section model from the study. However, the reviewer raises an excellent point with respect to the use of a positive control to compare the saccharin hydrogel to. To this end we have now repeated the biofilm assays including a pharmaceutical standard silver hydrogel wound dressing (Aquacell Ag⁺). These results demonstrated that the saccharin loaded hydrogels were more effective against *A. baumannii* biofilms (Figure 10A). We thank the reviewer again for this excellent suggestion which has no doubt improved the manuscript.

18th Feb 2025

Dear Prof. McCarthy,

Thank you for the submission of your manuscript to EMBO Molecular Medicine. We have now received feedback from the two reviewers who agreed to evaluate your manuscript. As you will see from their reports pasted below both referees support publication of your manuscript. Therefore, I am pleased to inform you that we will be able to accept your manuscript pending the following final amendments:

- 1) We note that some panels might be reused e.g. Fig. 9B and Fig S11. Please cite in the respective figure legend every reused panel.
- 2) Figures: Please remove figures from the main manuscript file and move their legends to the end of the file. Please rationalize the number of figures. We note that many figures have only 2-3 panels, e.g. Fig. 1 and 2 could be merged etc. Please check "Author Guidelines" for more information:
<https://www.embopress.org/page/journal/17574684/authorguide#figureformat>
<https://www.embopress.org/page/journal/17574684/authorguide#expandedview>
- 3) Author checklist: Please submit a complete checklist. <https://www.embopress.org/pb-assets/embosite/EMBO%20Press%20Author%20Checklist-1642513524327.xlsx>
- 4) In the main manuscript file, please do the following:
 - Please address all comments suggested by our data editors listed below:
 - o Data availability section:
 1. Please note that the specific URLs for GSE276752, GSE238183, 3DPX-020380 datasets are not provided in the data availability statement.
 - o Figure legends:
 1. Please note that the exact p values are not provided in the legends of figures 2B, 3B, D; 4A, 5A-E; 6A-D; 7A, B; 8B, 9C, 10A, B;
 2. Please indicate the statistical test used for data analysis in the legends of figures 8A
 3. Please note that in figures 6A-D there is a mismatch between the annotated p values in the figure legend and the annotated p values in the figure file that should be corrected.
 4. Please note that information related to n is missing in the legend of figure 8A
 5. Please note that the white arrows are not defined in the legend of figure 1C. This needs to be rectified.
 - Please provide corr. author information on the title page.
 - Add up to 5 keywords.
 - Add callout for Fig. 5A-D.
 - In Methods, statistical paragraph should reflect all information that you have filled in the Authors Checklist, especially regarding randomization, blinding, replication etc.
 - Indicate in legends exact n and exact p values, not a range, along with the statistical test used. To keep the figures "clear" some authors found providing an Appendix table Sx with all exact p-values preferable. You are welcome to do this if you want to.
 - Please include structured Methods section that includes a Reagents and Tools Table (should be uploaded as a separate file) followed by a Methods and Protocols section. More information on how to adhere to this format as well as downloadable templates (.docx) for the Reagents and Tools Table can be found in our author guidelines:
<https://www.embopress.org/page/journal/17574684/authorguide#structuredmethods>
An example of a paper with Structured Methods can be found here:
<https://www.embopress.org/doi/full/10.1038/s44320-024-00037-6#sec-4>
 - Rename "Competing interest" to "Disclosure Statement & Competing Interests". We updated our journal's competing interests policy in January 2022 and request authors to consider both actual and perceived competing interests. Please review the policy <https://www.embopress.org/competing-interests> and update your competing interests if necessary.
 - Data availability: Please use the following format to report the accession number of your data:

[data type]: [full name of the resource] [accession number/identifier] ([doi or URL or identifiers.org/DATABASE:ACCESSION])

Please check "Author Guidelines" for more information.

<https://www.embopress.org/page/journal/17574684/authorguide#availabilityofpublishedmaterial>

- Correct the reference citation in the reference list. Citations should be listed in alphabetical order. Where there are more than 10 authors on a paper, 10 will be listed, followed by "et al.". Remove DOIs and PMID/PMCID numbers. Please check "Author Guidelines" for more information.

<https://www.embopress.org/page/journal/17574684/authorguide#referencesformat>

5) Tables: Please rename all tables to Dataset EV1 etc. and add their legends to corresponding excel file as a separate tab/sheet. Update their callouts in the main manuscript file.

6) Movies: Please rename movie files to Movie EV1 etc. and zip each file with the corresponding legend that should be provided as a readme.txt file. Please update their callouts in main manuscript file.

7) Appendix: Please rename "Supplementary Figures" to "Appendix" and move all supplementary methods to the main Methods section. Rename supplementary figures to "Appendix Figure S1" etc. and update their callouts in the main manuscript file. Add table of content with page numbers on the title page. Remove all table and movie legends.

8) Funding: Please make sure that information about all sources of funding are complete in both our submission system and in the manuscript. Currently, Research Mobility Award from the Young European Research University Network, Research Grants BB/T007168/1 and BB/T006625-1 from the Biotechnology and Biological Sciences Research Council, BBSRC research grant BB/W000393/1, a grant by the North Norwegian Health Trust (HelseNord, HN 1688-23), are missing in our submission system. Please correct.

9) The Paper Explained: Please provide "The Paper Explained" and add it to the main manuscript text. Please check "Author Guidelines" for more information. <https://www.embopress.org/page/journal/17574684/authorguide#researcharticleguide>

10) Synopsis: Every published paper now includes a 'Synopsis' to further enhance discoverability. Synopses are displayed on the journal webpage and are freely accessible to all readers. They include separate synopsis image and synopsis text.

- Synopsis image: Please provide a striking image or visual abstract as a high-resolution jpeg file 550 px-wide x (300-600)-px high to illustrate your article.

- Synopsis text: Please provide a short standfirst (maximum of 300 characters, including space) as well as 2-5 one sentence bullet points that summarise the paper as a .doc file. Please write the bullet points to summarise the key NEW findings. They should be designed to be complementary to the abstract - i.e. not repeat the same text. We encourage inclusion of key acronyms and quantitative information (maximum of 30 words / bullet point). Please use the passive voice.

11) As part of the EMBO Publications transparent editorial process initiative (see our Editorial at <http://embomolmed.embopress.org/content/2/9/329>), EMBO Molecular Medicine will publish online a Review Process File (RPF) to accompany accepted manuscripts. This file will be published in conjunction with your paper and will include the anonymous referee reports, your point-by-point response and all pertinent correspondence relating to the manuscript. Let us know whether you agree with the publication of the RPF and as here, if you want to remove or not any figures from it prior to publication. Please note that the Authors checklist will be published at the end of the RPF.

12) Please provide a point-by-point letter INCLUDING my comments as well as the reviewer's reports and your detailed responses (as Word file).

I look forward to reading a new revised version of your manuscript as soon as possible.

Yours sincerely,

Zeljko Durdevic

*** Instructions to submit your revised manuscript ***

1) a .docx formatted version of the manuscript text (including Figure legends and tables)

2) Separate figure files*

3) supplemental information as Expanded View and/or Appendix. Please carefully check the authors guidelines for formatting Expanded view and Appendix figures and tables at <https://www.embopress.org/page/journal/17574684/authorguide#expandedview>

4) a letter INCLUDING the reviewer's reports and your detailed responses to their comments (as Word file).

5) The paper explained: EMBO Molecular Medicine articles are accompanied by a summary of the articles to emphasize the major findings in the paper and their medical implications for the non-specialist reader. Please provide a draft summary of your article highlighting

6) Author contributions: the contribution of every author must be detailed in a separate section.

7) EMBO Molecular Medicine now requires a complete author checklist (<https://www.embopress.org/page/journal/17574684/authorguide>) to be submitted with all revised manuscripts. Please use the checklist as guideline for the sort of information we need WITHIN the manuscript. The checklist should only be filled with page numbers where the information can be found. This is particularly important for animal reporting, antibody dilutions (missing) and exact values and n that should be indicated instead of a range.

8) Every published paper now includes a 'Synopsis' to further enhance discoverability. Synopses are displayed on the journal webpage and are freely accessible to all readers. They include a short stand first (maximum of 300 characters, including space) as well as 2-5 one sentence bullet points that summarise the paper. Please write the bullet points to summarise the key NEW findings. They should be designed to be complementary to the abstract - i.e. not repeat the same text. We encourage inclusion of key acronyms and quantitative information (maximum of 30 words / bullet point). Please use the passive voice. Please attach these in a separate file or send them by email, we will incorporate them accordingly.

You are also welcome to suggest a striking image or visual abstract to illustrate your article. If you do please provide a jpeg file 550 px-wide x 300-600px high.

9) A Conflict of Interest statement should be provided in the main text

10) Please note that we now mandate that all corresponding authors list an ORCID digital identifier. This takes <90 seconds to complete. We encourage all authors to supply an ORCID identifier, which will be linked to their name for unambiguous name identification.

Currently, our records indicate that the ORCID for your account is 0000-0002-7480-6352.

Link Not Available

11) Include a Reagents and Tools Table as part of the Methods section, which can be downloaded from our author guidelines (<https://www.embopress.org/page/journal/17574684/authorguide#structuredmethods>)

Photos 400-800 DPI

*Additional important information regarding figures and illustrations can be found at

<https://bit.ly/EMBOPressFigurePreparationGuideline>. See also figure legend preparation guidelines:

<https://www.embopress.org/page/journal/17574684/authorguide#figureformat>

***** Reviewer's comments *****

Referee #1 (Remarks for Author):

All my questions have been addressed

Referee #2 (Remarks for Author):

I thank the authors for their efforts to address all of my comments on the previous version of the manuscript. The conclusions in this version of the manuscript are now well supported by the data and the authors have addressed all of my concerns.

The authors addressed the remaining editorial issues.

6th Mar 2025

Dear Prof. McCarthy,

We are pleased to inform you that your manuscript is accepted for publication and is now being sent to our publisher to be included in the next available issue of EMBO Molecular Medicine.
